# Fantastic Flails and Where to Find Them: The Body of Evidence for the Existence of Flails in the Early and High Medieval Eras in Western, Central, and Southern Europe

**Alistair F. Holdsworth** [1,2]

1   Independent Researcher, Manchester M1 7HF, UK; aliholdsworth@hotmail.com or
    alistair.holdsworth@manchester.ac.uk
2   Department of Chemical Engineering, The University of Manchester, Oxford Road, Manchester M13 9PL, UK

**Abstract:** Flails are one of the most contentious and misunderstood classes of medieval weaponry, despite their prevalence in popular media: some researchers question their existence entirely and the bulk of historians are skeptical of widespread temporal and geographical prevalence, while others, and a significant volume of period evidence, would argue the contrary. While the expansive use of flails in Eastern Europe and Byzantium is familiar, many Central, Western, and Southern European sources are less well known or largely forgotten, especially those stemming from the later-early and early high medieval eras (up to 1250). In this work, I collate and discuss the bulk of the available literary references and artistic depictions of flails and their use alongside some of the archaeological finds from Western, Central, and Southern Europe, with an emphasis on the 12th and 13th centuries. The significance of this volume of evidence is examined, and an assessment of flails as a part of medieval culture and warfare is considered. Collectively, this would suggest that knowledge of flails as instruments of war and associated cultural connotations, if not their actual prevalence and use in warfare, was far more widespread across Europe this time period than has been previously estimated.

**Keywords:** flail; early medieval; high medieval; flexible weapon; ball-and-chain; morningstar; war flail; holy water sprinkler; threshing flail; improvised weapon



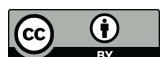

## 1. Introduction

Flails, which for the purposes of this work are defined as a weapon consisting of one or more solid striking heads attached to a one- or two-handed grip via flexible links of chain, cord, or similar, are one of the most contentious and misunderstood aspects of medieval warfare (Lepage 2005; Tzouriadis 2017). Both academic and amateur historians display a spread of opinions on the provenance and proliferation of flails, especially when considering dates up to the mid- or even late 13th century in Western, Central, and Southern Europe, despite the widespread appearance in modern media such as film, videogames, and "popular history" writings (Sturtevant 2016a, 2017; Manning 2016; Fernandes 2019; Dahm 2018; Tardy 2020; Moreno 2015; Devereaux 2019; Mondschein 2017; Tunis 1999; Dougherty 2008). The depictions of flails, especially in the form of illuminated manuscript folios and even surviving examples, are commonly acknowledged to increase throughout the 14th century and beyond.

The opinions on flails range from complete dismissal or belief that nearly all provenances are fallacy at one extreme (Sturtevant 2016a, 2016b, 2017; Warner 1968; O'Bryan 2013; Brooks 2019), via indifference or overstated use (Manning 2016; Oakeshott 2012), to a general acceptance of period knowledge and at least sporadic utilisation in war at the other end of the spectrum (Gallardo 2016; Moreno 2015; Nilsson 2021; Campagnano 2015).

I believe the former of these viewpoints, and to a degree the second, arise from a poor awareness of period sources, combined with an at-times poorly researched, sensationalist, and occasionally inflammatory approaches to "modern (and popular) medievalism"

(Tzouriadis 2017; Utz 2016; Gallardo 2016). A great many artistic and literary sources from the 11th through 13th centuries depict or describe the use of flails in battle; the majority of these are publicly available if not particularly easy to access or well known but many, including some more obscure sources, were extensively described by the historians of the 19th century (Viollet-le-Duc 1874; Sternberg 1886; Oman 1898), and so their omission from recent publications in the present data age can be probably be attributed to poor scholarship, ingrained prejudices, or logical fallacies. Nonetheless, to my knowledge, no research has collated and analysed the available 11th through 13th century sources from Western, Central, and Southern Europe depicting and describing flails, and as such to address this gap in the literature, I have gathered those known to me and presented them here with discussion to raise awareness of these earlier, more poorly known examples, the likely prevalence and use of flails more generally, and the possible routes by which the technology migrated with some possible origins, cultural context, and associated symbology. These primary sources encompass a combination of artistic representations (statues, carvings, frescoes, manuscript illuminations, etc.), literary references, and archaeological finds (DeVries and Smith 2012; Nicolle 1999). Several relevant sources from outside the area of interest are highlighted for context where appropriate to the discussion.

For the purposes of this work, I will consider four categories of flail which are depicted or described in primary sources: threshing flails—farming tools used for millennia but occasionally pressed into service as improvised weapons; weaponised threshing flails—weaponised derivatives of the former which became popular in Europe during and after the 14th century; scourges—sometimes incorrectly referred to as flails, which are punishment devices rather than weapons, often depicted in religious settings and referenced heavily in the Bible (Moreno 2015); and war flails—a broad range of weapons including the "ball-and chain" type, which collectively are the primary source of contention around this class of weaponry and the primary focus of this work. Examples of each of these are presented in Figure 1.

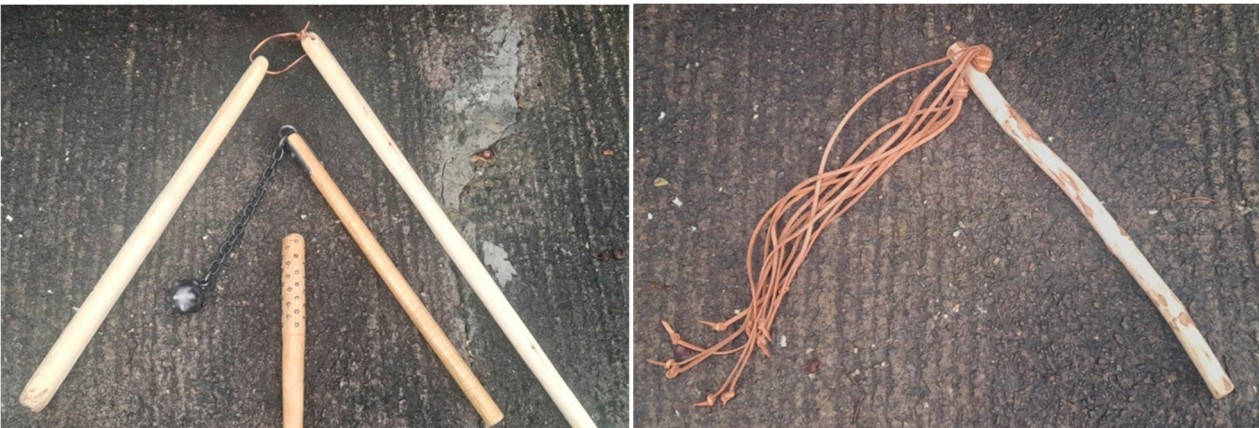

**Figure 1.** Examples of threshing flail (**top**), war flail (**middle**), and warclub (demonstrative of the pattern weaponisation of threshing flails) produced by and from the collection of the author (**left**, war flail ironwork by Wieland Forge), and scourge (**right**) kindly loaned from the collection of M. J. Dandy. Photos: © the author. Note that these are "combat-safe" interpretations based on period depictions.

I believe the collection of primary and secondary sources presented here is extensive but is by no means complete, and thus should not be considered exhaustive. Please note that while parallel development of flexible weaponry occurred during the medieval period across large parts of the world including Japan (Gonzalez 2013), China (Needham 1959; Faulliot 1982), Korea (Herrera 2014), the pre-colonial Americas (Courville 1948), and many other places, (Costa 2015; Moreno 2015; Gallardo 2016; Escobar 2018) these are beyond the scope of this work and will not be discussed further unless pertinent. The aim of this work does not include the development of weapon typologies but rather to provide a high-level

overview and to highlight the previously underrated prevalence of flails within the region and period of interest by reviewing the available sources. A brief discussion of the likely historical use of flails in warfare will be presented, alongside some conclusions regarding the possible routes by which flails were introduced to Western, Central, and Southern Europe, likely before, but most certainly within the early medieval period; it was theorised that war flails migrated westwards from Eastern Europe during the 10th–12th centuries, but the provenance of many primary sources refutes this concept.

Much of the skepticism surrounding flails arise from the challenges in controlling such flexible weapons practically, with some researchers describing their motions as chaotic and as harmful to the wielder and their allies as to the enemy. (Sturtevant 2016a; Brooks 2019; Moreno 2015). As a riposte to this, several late Saxon (10th–11th century) sources describe the ability of farmers to thresh crops without injury (Hill 1998; Gallardo 2016), and modern reproductions from 15th–16th century "fight manuals" comprehensively demonstrate the opposite: with appropriate training and practice, flails are as controllable and precise as other weapons (Rüther 2023). The advantage of flails in their ability to "wrap around" shields, limbs, and weapons (Zábojník 2009), and hit at least as hard, if not harder, than rigid weapons of similar weight and design (e.g., maces) with significant blunt force (Rüther 2022; Van Dyke et al. 2007), makes them hard to block (Warner 1968; Moreno 2015), effective against certain types of armour (chainmail), and especially dangerous under some circumstances. The "safe" use of flails in controlled environments, such as historical re-enactments, is a challenging balance to strike—maintaining the authenticity while also not risking serious injury to participants and observers alike (Radtchenko 2006).

## 2. Flails in Antiquity

The earliest historical depictions or descriptions of flails appear as symbols of kingly authority in both ancient Egypt and Mesopotamia (Civil 1963; Moreno 2015), between 3500 and 2500 BC. In the case of the former, this was paired with the shepherd's crook by the Second Dynasty (2890–2696 BC), both attributed to the deity Osiris. Surviving examples are present in the sarcophagus of Tutankhamen (Figure 2), the crook representing kingship, and the flail (or possibly scourge) the fertility of the land (or the law and justice embodied in the Pharaoh's authority) (Steele 2002; Norris 2015), or alternatively the ability to provide both grain and meat (Costa 2015); both of these symbols are utilised as hieroglyphs. This demonstrates the basis for flails as farming tools at least as far back as the third and maybe fourth millennium BC, with the associated symbology emphasising the divine right and duty of the Pharaohs.

The earliest depictions of war flails, resembling medieval threshing flails in design, appear on a series of coins from Thessaly (e.g., Ancient Greece, in Figure 3), dating from approximately 400 BC (Robinson 1933), although coins from the same time period also present flails as tools as well, likely mirroring their earlier symbology of the royal right to rule (Seltman 1946). A comprehensive selection of such artefacts can be found here (https://www.coinarchives.com/a/results.php?search=flail, accessed on 25 October 2023), where some Roman gladiators (2nd century BC coins) are also depicted utilising flails (Markowitz 2021; Courville 1948). Furthermore, the Roman writer Publius (or Vegetius, late 4th century) describes the use of "Plumbata" by cavalry and infantry—interpreted to be a lead or iron ball connected with a line to a wooden handle (with a doubled meaning for lead-weighted darts) in his writing *De Re Militari* (Schultz 1889; Carley 1962), a text that was known and translated in the early part of the high medieval era (Allmand 2011), and will be discussed later. The prevalence of depictions flails on coins from Thessaly, whose cavalry were one of the cornerstones of Alexander the Great's battlefield success, would suggest that these were employed at least to some degree during the ancient period alongside the more common spear, as one summary of such finds describes the occurrence of flails as being at least as common as swords (Papaioannou 2019).

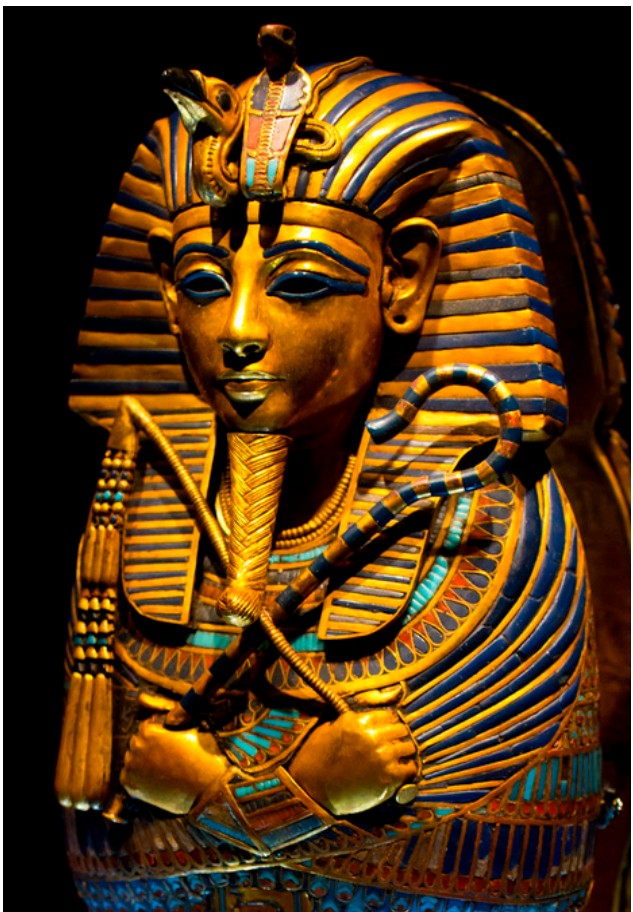

**Figure 2.** The coffinette of Tutankhamen (Pacific Science Centre, Seattle, WA, USA) depicting the crook and flail. (© D. Denisenkov, permission granted for non-commercial reproduction under CC BY-SA 2.0 DEED license, image cropped from original (https://www.flickr.com/photos/ddenisen/7364444958/in/photostream/, accessed on 25 October 2023) but otherwise unaltered).

Given the rapid expansion and ultimate expanse of Alexander the Great's Macedonian Empire at its height, which ranged from the Balkans in the northwest, as far south as Egypt and as far east as India, it would not be unfeasible for flails to have spread along with the members of his army who settled in the newly captured provinces.

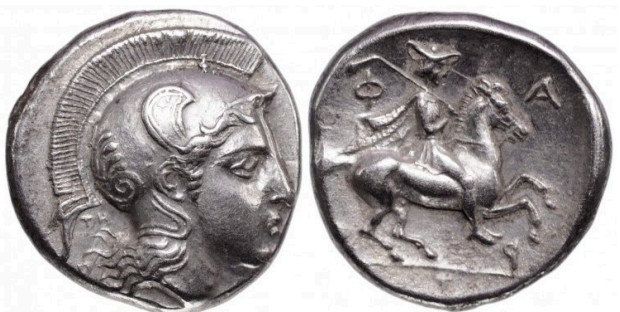

**Figure 3.** Thessalian coin (400–370 BC) depicting cavalryman with flail. (© Forum Ancient Coins (www.forumancientcoins.com), permission granted for non-commercial reproduction, exact image from https://www.forumancientcoins.com/gallery/displayimage.php?pid=155876, accessed on 25 October 2023).

Adoption by subjugated peoples within the Macedonian successor states including, perhaps, in the more northerly extremes their territory bordering the central Asian steppe,

the ancestors of the later Hunnic invaders of Europe and other such steppe peoples could then have occurred, though independent invention cannot be discounted. Archaeological finds from the Alps indicate that perhaps even the Celtic peoples fought by the Romans possessed knowledge of flails or perhaps maces (Rageth and Zanier 2010). Items resembling flail heads and skeletal remains bearing injuries consistent with blunt force trauma from these have been found in ancient Sarmatia (modern-day Ukraine and surrounding areas, dated 3rd–4th century BC) (Kotowicz 2006; Mamontov 2018). Ornately carved stone balls found in Scotland and dating from ancient British times have been postulated to be "flail heads", in a similar manner to the bolas throwing weapons of Central and Southern America (Smith 1876; Cowper 1906; Courville 1948; O'Bryan 2013).

### 3. Cultural and Literary Context of Flails in the Medieval Era

At this stage, it is pertinent to discuss the various historical and more recent translations of the overarching term *flail* into European languages (Snook 1979; Tihle 2017), as even in modern parlance there is significant confusion and no uniformity in the many and varied terms used to describe such weapons in primary (Carley 1962) and secondary sources (Tzouriadis 2017). In modern English, the flail as a weapon is sometimes referred to interchangeably as the *war flail*, *morning star*, or *holy water sprinkler*, while in Old English, the term *fleel* or *flael* is used. In modern French, the common term is *fléau d'armes* (war flail), while in older texts, the term *plomée* or *plommée* is commonly utilised, though these have several meanings which will be discussed later; a few other less common terms will be highlighted as necessary, given the prevalence of primary French language sources. Modern German has several more descriptive terms, including *Kriegsflegel* (war flail), *Morgenstern* (morning star), *Kettenmorgenstern* (chain morning star), *Streitflegel* (lit. fight flail), and *Schlachtgeissel* (battle whip). In most Slavic languages (Polish, Ukrainian, Russian), the term *kisten* or *kiścień* is used, with *Řemdih* and *Remdik* used in Czech and Slovak respectively. Modern Spanish and Portuguese use the term *mangual*, while Italian is *mazzafrusto*, with the period term *flagellum* and many other more specific derivatives (Nicolotti 2017).

The semantics of the term *flail* begin to become complex around the time of the Hunnic invasion of the 4th–5th century AD. The high medieval (1160–1170) poem *Servaes Legende*, attested to Heinrich von Veldeke (d. after 1184), based upon the previous *Gesta Sancti Servatii* (c. 1126), which found its roots in the *Historia Francorum* (by Gregory of Tours, d. 594), describes the following exchange between Attila the Hun and the city of Troyes (drawn from the incomplete translation by Bathgate, Line 89 (Bathgate 1990):

> "Then Troyes: Bishop Lupus asks Atilla (Attila), "why do you destroy the whole land?"
>
> Atilla (Attila) says, "I am God's flail; open the gates, and I will thresh both woman and man!"

In this context, *the flail of God* Attila refers himself as being can be misconstrued due to a tendency of the meaning between *flail* and *scourge* to merge, though the reference to threshing means this requires somewhat deeper analysis. This merged meaning of flail and scourge arises from the Latin term *Flagella*, meaning flail (several types), and also scourge (the whip-type devices) to refer to (Nicolotti 2017), in the latter case, a scourge in its alternate definition as a source of great suffering or trouble. Similar observations are seen in French with the term *fléau*, where alone, this word refers to *scourge*, but with the suffix *d'armes*, refers to the weapons described here (Havard 1876). The evolution of language incorporated the further meanings of the word *flail*, at least in English, from just the farming tool to the associated military definition primarily discussed here (Izdebska 2016); the chaotic *flailing* of threshing flails is indeed the origin of the modern adjective to describe wild circular motions (such as of the arms), representing a further branching of the semantics of the word (Hill 1998).

The epithet *Flagellum Dei*, meaning scourge/flail of God, was first loosely associated with Attila around the turn of the 7th century, but had acquired a truly negative connotation referring to him by the 11th century, at least within Western Europe; the Eastern Roman

Empire (centred on Byzantium) held a much higher view of the *Flagellum Dei* (Ridley 1992). The Flagellum Dei epithet was also applied to the Lombards at times during and after their conquest of northern Italy (568–774), to the Hungarians (Magyars) (Williams 1979) during their various invasions of Germany during the 10th century (Köstler 1883), and also the Magyars and Pechenegs by the Byzantines during the 10th–12th centuries (Paron 2021). Even certain Christians have been described as *Flagellum Dei* or rather *fléau de Dieu*—in reference to punishing the impiety of the Flemmings and English (under King John) at the battle of Bouvines (1214) (Leon 2019). The late 14th century French work *L'Arbe des Batailles* criticises violence as a scourge upon the land, but also recognises it as a punishment from God, who allows it to occur as a price for sins; the soldiers who provide the violence are, in this context, the *flail of God* or *les executeurs de nostre Seigneur* (Niewiński 2019). The 15th century French nobleman and general Jean de Dueil was labelled *le fléau des Anglais* (plague/scourge of the English) during the Hundred Years War (de Boislisle 1943), and the English solider Richard Bingham (1528–1599) was referred to by the epithet *the Flail of Connaught* for his oppressive actions against the Irish (Herron 2002). Indeed, the term *Flagellum Dei* could be applied to "the enemy generally, regardless of their origin (Allmand 1999)".

In a reversal of meaning from ancient times, by the early parts of the high medieval period (as attested in the *Domesday Book*), the flail had become a symbol of poverty and villeinhood (Stoljar 1985; Homans 1941; Power 1940; Hyams 1970), remaining a symbol of servitude and menial (farming) labour well into the Renaissance (Beauchamp 1981), although these labours were viewed in something of a positive light in the period (McDougall 1983), flailing being viewed as a metaphor for God's struggles inflicted upon Christians to test the strength of their belief (Lipton 1999). To break from one's designated place in society was at times seen as a threat to one's betters, as the French epic poem *Partonopeus de Blois* (c. 1170–1180) describes—drawing upon the symbols of "plough, pitchfork, and flail" as those of servitude, warning against the perils of a *fils à villain* (lit. "naughty son") being raised from the dirt to those of higher office and political influence, directly wishing upon them significant misfortune (Eley 2011). This symbolism was further carried into the proto-science fiction works of the English poet Edmund Spenser (1552–1599), who incorporated a flail into an "iron man" character named Telus into several of his works; the flail included as a symbol of servitude to his human masters (McCulloch 2011).

From the 13th and into the 14th centuries, flails begin appearing in heraldic designs in Germany, such as those of Pflegelberg (Erisman n.d.; Frutos 2019; Moreno 2015), with similar depictions (a lion holding a flail) noted in Aragaonese arms (Clemmensen 2013; Moreno 2015).

## 4. Depictions and Descriptions of Threshing Flails as Improvised Weapons in Western, Central, and Southern Europe in the Early and High Medieval Eras

Threshing flails are ancient farming tools used to separate grains from stalks during harvesting, with their use in parts of Europe dating back at least two millennia (Andrén 2005), the latest occurrence being in parts of Scandinavia in about 1000 AD (Myrdal 1997). As symbols of the peasantry, along with the pitchfork and the scythe, threshing flails are more tools than implements of warfare, but when push comes to shove, such implements can be turned on their fellow man as improvised weapons, a practice which is generally accepted by historians (Moreno 2015; Gallardo 2016). The Greek philosopher and mathematician Euclid is reported to have considered it "a futile labour [to teach mathematics to those who toil with] flail and axe", indicating that the users of such tools were not worthy of being raised up intellectually (Gug 1986), and as such connotates these tools with the uneducated lower-classes.

In Irish custom, the haft of a threshing flail was usually made of ash, while the beater (swingle or souple) could either be made of ash, hazel, or holly (Mac Coitir 2020; Newman 1985; Doyle et al. n.d.). Various means exist of attaching the two portions of a threshing flail together to ensure sufficient flexibility of the head, with the exact design employed varying

with locale of manufacture (Newman 1985), the minutiae of which are beyond the scope of this work, though a selection of hinges is presented below (Figure 4, Veiga de Oliveira et al. 1983). The primary sources describing threshing flails as improvised weapons are now presented and discussed.

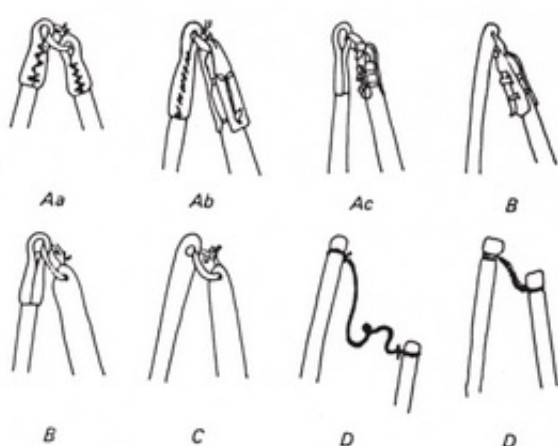

**Figure 4.** Variation in threshing flail hinges from Portuguese tradition: type A: double leather loops; type B: single leather loop; type C: bored wood; and type D: thong or rope hinge. (Reproduced with the kind permission of Etnografica Press, CRIA (Centre for Research in Anthropology), Lisbon, Portugal. From Ref: Veiga de Oliveira et al. 1983; sourced from Gallardo 2016).

The Gaelic folklore tradition describing the legend of Finn mac Cumaill (Fionn Mac Cumhaill) likely dates from before the high medieval period, but references to the death of this character and a particular scene described in hell are found from the 10th century, becoming a key part of the story by the 12th century. One of the other characters in the story (Oscar) is described as using a flail in hell to keep demons away, which is later transferred to the titular character in some variations of the story (Macculloch 1916). The flail described is "like that used for threshing grain", but via divine favour (via an angel), the flail strap is made so that "it will last"—and made more durable (Maher 2018; O'Hogain 1986). This provides an interesting example of a threshing flail being used in a folklore setting as an improvised weapon and highlights that perhaps the hinges were prone to breakage. The modern English translation from the original Celtic by Maher is incomplete, but analyses this section in detail (Maher 2018).

In the 19th century writing *The History of Normandy and of England*, Palgrave describes the "proud Norman peasantry" rising up and mustering with an assortment of weapons including (threshing) flails to fight the English (Saxons?) and Germans invading Normandy during the time of Duke Richard II (r. 996–1026) and later regents, during battles at Bihorel and Maromme and on the Contentin peninsula (near Cherbourg), though period sources for these references are not provided (Palgrave 1864).

A 12th century fresco in the Royal Pantheon of San Isidoro (Spain) depicts peasant militias (described as pawns from the town of Bernesga, Spain, in modern references) armed with threshing flails supporting noble cavalrymen in battle (Moreno 2015; Gallardo 2016).

The Welsh tale *Culhwch and Olwen* is connected to the legend of King Arthur and his warriors and believed to originate orally from the 11th or 12th century, though only manuscripts from the early 14th and 15th centuries survive. The physical strength of one of the characters (Cacamwri, a servant of Arthur) is demonstrated by rending a barn to dust with a threshing flail. While allegorical and most certainly prone to the author's exaggerations, this does highlight the period knowledge of the potential of a flail as a tool of destruction (Toffee 2013). A complete translation of the original tale (with notes) is available at https://www.culhwch.info/, accessed 25 October 2023.

An early 12th century illuminated manuscript from Citeaux, an illustrated version of the earlier *Moralia in Job* (Dijon BM MS.173, Figure 5), depicts a folio (fol. 148r) of a

threshing flail in what is described by Thompson as "a humorous play on the flail of God (*Flagellum Dei*) described in the book" (Thompson 2000). This work is associated with Pope Gregory the Great (d. c. 604), of which many reproductions were produced over the subsequent centuries.

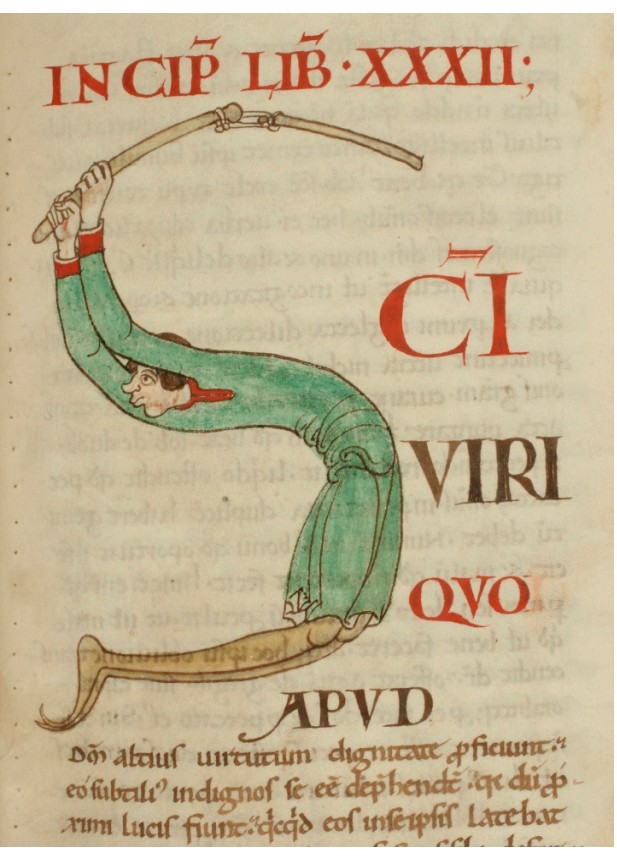

**Figure 5.** Fol. 148r from *Citeaux Moralia in Job* (© Bibliothèque Municipale de Dijon, MS. 173, permission granted for non-commercial reproduction).

The figure and the flail form a contorted "S" shape as an enlarged dropped first letter of the page and represent one of the better depictions of the "hinge" of a threshing flail in period illustrations. In the context described in her thesis (Thompson 2000), based partly upon the interpretation of the work of Rudolph (Rudolph 1997), Thompson postulates that the folio represents the "spiritual struggle", and as such this can be likened to the use of the flail as a weapon to ward off temptation (Thompson 2000). This may have had a greater connection to the working classes intended by the author of the manuscript, but this is open to speculation.

In his mid-12th century chronicle *Roman de Rou*, the poet Wace described the mustering of the Saxon peasants (fyrd) for the Battle of Hastings with (pitch)forks amongst other weapons and tools they had to hand (Planché 1876). Here, the word "tinels" is used—referring to "heavy wooden implements"—perhaps clubs or flails, as described in the full-text translation by Taylor (Taylor 1837).

"Li vilains des viles aplouent

Tels armes portent com ils trouvcnt;

Machus portent e grans pels

Forches ferrdes e tinels."

"The villeins of the towns applaud (gather?)

The arms bear where they find

March forth. . .?

Forks of iron and [heavy wooden implements]."

During the Anarchy (1138–1153), the English war of succession that followed the death of Henry I (d. 1135), a power struggle between two rival strands of the royal family resulted in numerous campaigns across England and modern-day France. One secondary source attests to, as with the later description of Jordan Fantosme (see below), the "peasants and churls turning out and driving the hated Guirribecs back over the border with fork and flail" (Green 1888), though no original period reference is provided to corroborate this. In this context, "Guirribec" was used a derogatory term for the Angevins, who were likely suffering from dysentery (Bradbury 1990).

*Van den vos Reinaerde* (the tale of Reynard the Fox in Middle Dutch) is a mid-12th century German tale (c. 1250 for the Dutch translation) telling the story of anthropomorphic animals in a manner comprising both human and animal behaviours. In the tale, a bear is harassing a village, which rouses the characters to gather their arms, a (threshing) flail (vhlegel) and pitchfork included, to kill the bear (Bouwman and Besamusca 2009):

"Sulc was die eenen bessem brochte,

sulc eenen vleghel, sulc een rake,

sulc quam gheloepen met eenen stake."

One of the earliest primary references I am aware of where threshing flails are used as improvised weapons outside of a fictional setting comes from The *Chronicle of the War between the English and the Scots in 1173 and 1174* by Jordan Fantosme (Hosler 2017; Day 2010; Short 2022). He describes their use against Flemish mercenaries alongside (pitch)forks (the extended modern translation is provided for context):

"N'i aveit el païs ne villain ne corbel/N'alast Flamens destruire a furke e a fleel"

"In all the countryside there as neither villein nor peasant who did not go after the Flemings with fork and flail to destroy them. . . . . .by fifteen, by forties, by hundreds and by thousands/by main force they make them tumble into the ditches . . . Upon their bodies descend crows and buzzards/who carry away the souls to the fire which ever burns'"

In this context, given the specific reference to the villains and the known symbology of "fork and flail" as those of servitude and the lower classes, he must be referring to threshing flails as tools of the peasantry. Matthew Paris repeats this tale in his *Historia Anglorum* (c. 1250–1259) (Oman 1898).

The late 12th century French romance *Perceval ou le Conte du Graal* (Perceval and the Story of the Grail) is an Arthurian tale written by Chretien de Troyes (fl. 1160–1191) which describes the story of the titular character on the quest for the holy grail (not the cup as per modern adaptions of the story). In the story, Gawain is accused of murder, causing (to quote McDowell) "every single rogue to snatch up a pitchfork or a hammer or a flail" (Lines 5945–5946), the mob threatening to tear apart the tower of Gawain (McDowell 2007). In this context, the flails described would be threshing flails, being tools used as improvised weapons. This was drawn from the complete translation of the original (Old French) text into English by Raffel (Raffel 1999). It seems that threshing flails were not only employed as occasional improvised weapons but at times as implements of domestic violence—in the early 13th century, a Hereford bailiff by the name of Robert Praepositus "struck his daughter in the eye with a flail and the eye was lost". She survived the ordeal and went on to marry (Hillaby 2004). A similar court entry was recorded in Shropshire in 1256 (Skinner 2017).

Fol. 59v from the *Chronicle of the Fifth Crusade* (Corpus Christi MS 016II, Chronica Maiora II, Figure 6) by Matthew Paris (d. 1259), dated to the mid-13th century, depicts the Frisian Hayo von Wolvega attacking the Tower of Damietta (modern-day Egypt) with a threshing flail as an improvised weapon during the FIfth Crusade (1217–1221) (Day 2010; James 1925–1926; Hewitt 1855; Moreno 2015). This illumination likely stems from

the testimony of Oliver of Paderborn (c. 1170–1227), who accompanied the FIfth Crusade to the Holy Land. In his *Historia Damiatina*, "a flail by which grain is usually threshed" was weaponised by "interweaving it" with chains. A full version or translation of this text is unavailable to me. Whilst this feature is omitted from the folio below (Edwards 2019; Edbury 2016), it is a strong candidate for early "weaponising" of threshing flails as is discussed later (see also Figure 1).

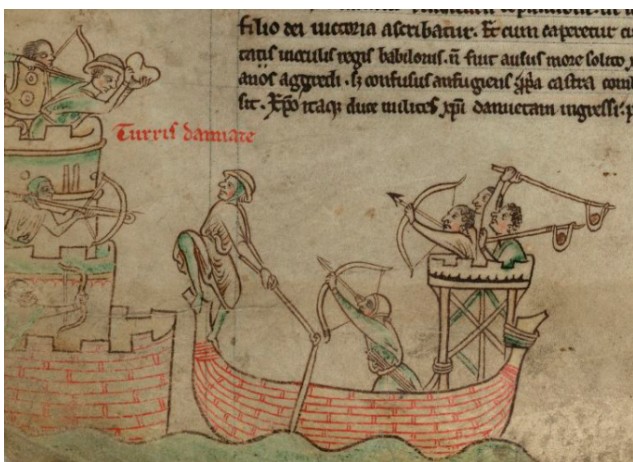

**Figure 6.** Fol. 59v from the Chronicle of the Fifth Crusade (also known as Chromica Maiora II, © The Parker Library, Corpus Christi College, Cambridge, MS 016II, permission granted for non-commercial reproduction under CC BY-NC 4.0 license, image from https://parker.stanford.edu/parker/catalog/canvas-008898466b1b9f0019740bcf8093af5c, accessed on 25 October 2023). Note also the staff slingers and archers also in the boat.

Improvised threshing flails were still in use later in the medieval period (13th–14th centuries) by the entrenched lower classes as well, especially in resistance to expansionist tendencies of land-hungry nobility of differing ethnicities (Zientara 1970), even predating the use of weaponised-type threshing flails during the Hussite Wars (1419–1434) by at least a century (Moreno 2015; Wasiak 2012; Tihle 2017).

One of the more interesting folios from the early 14th century (c. 1330), the manuscript *The Taymouth Hours* (BL Yates Thompson MS13, f185v, Figure 7) depicts a shortened threshing flail in the hands of a half-man half-lion (or other big cat) also carrying a (conical wicker?) shield with a gigue strap (British Library Catalogue n.d.). To my knowledge, this is the only depiction of this kind of flail within early high medieval sources.

In the Celtic folk tale of Irish or Scottish origin entitled *The Battle of the Birds*, the birds in the story are described as using flails as weapons to fight the mice, with the more modern, translated text specifically mentioning threshing flails being utilised as improvised weapons, albeit in a fantastical setting. The story and its variants were first recorded in the 15th century but is believed to be much older in origin, due to the prevalence of similar tales across Europe and into Asia (Jacobs 1892). The work of Jacobs represents a complete, adapted translation of the story, as with the following example which comes from the same reference (Jacobs 1892).

The tale entitled *The Lad with the Goat Skin* tells the fable of a young smith of Dublin who, smitten with a princess, is bidden by the King of Dublin to collect a (threshing) flail from hell, which is said to be feared by the Danes harrying Ireland at the time (Kennedy 1866; Jacobs 1892). As with many folk tales, this piece is first referenced in the medieval era, but given the context of the tale, the story is likely a few centuries older, before the invasion of Ireland by the Normans in the 12th and 13th centuries (Kennedy 1866; Jacobs 1892). The overall structure of the story mirrors that of the c. 1200 French (possibly Breton) story *Lai d'Eliduc* (the Lays of Eliduc), though there are no specific references to flails in that piece that can be determined, although war is described as "*fléau de la guerre*"—lit. "the

scourge of war" (de France 1820). The 19th century illustrations accompanying this and the previous tale illustrate them as threshing flails (Jacobs 1892).

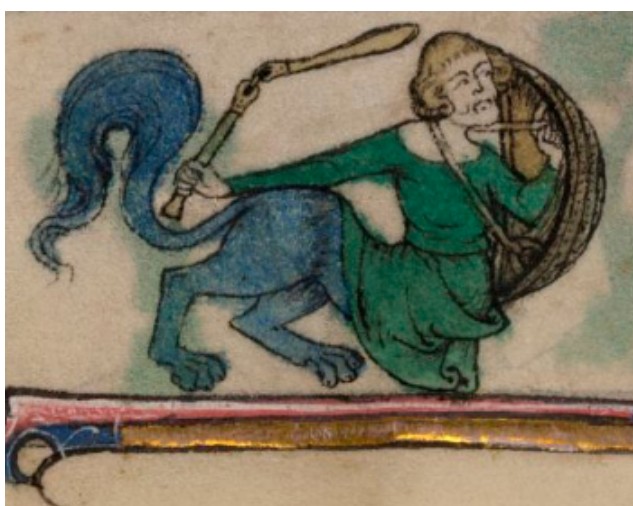

**Figure 7.** Fol. 185v from The Taymouth Hours. (© British Library Board, BL MS Yates Thompson 13, permission granted for non-commercial reproduction, image from https://imagesonline.bl.uk/asset/170871/, accessed on 25 October 2023).

## 5. The Weaponisation of Threshing Flails in the Later Medieval Period

Even the biggest sceptics of flails being used in warfare (Brooks 2019; Sturtevant 2016a, 2016b, 2017) cannot deny the volume of evidence for their provenance during the Hussite Wars (1419–1434). I will dub the type of flails used in this conflict as *weaponised threshing flails*, reflecting the similarity in design to the farming tool with adaption for greater lethality. Depictions of weapons evidently derived from threshing flails appear from around the turn of the 14th into the 15th centuries (Snook 1979; Tihle 2017; DeVries and Smith 2012; Sturtevant 2017; Grabarczyk 2000). This "weaponisation" appears to have been achieved in a similar manner to earlier warclubs (see folio from Moniage de Guillaume and Figure 1) (Tzouriadis and Deacon 2020; Tzouriadis 2017), via the addition of iron or copper nails as force concentrators, rather than relying purely on the bludgeoning effect of the wooden head, though more elaborate designs with all-metal heads were certainly produced, and perhaps used by the Mongols (Moreno 2015; Kuleshov 2019). As long, unwieldy, two-handed weapons, these are nearly always depicted being used by infantry, with several "fight manuals", such as works of the Germans Hans Talhoffer (MS Thott 290 2o, 1459; see https://wiktenauer.com/wiki/Talhoffer_Fechtbuch_(MS_Thott.290.2%C2%BA), accessed on 25 October 2023) (Hull 2007; Talaga and Ridgeway 2020; Aveyard et al. 2014; Grant 2020), and Paulus Hector Mair (mid-16th century, e.g., Amplissimum de Arte Athletica (Cod.icon. 393); see https://wiktenauer.com/wiki/Opus_Amplissimum_de_Arte_Athletica_(Cod.icon._393), accessed on 25 October 2023) (Impey 2019; Moreno 2015) and a number of Spanish sources (Ortiz 2016) detailing their use alongside a plethora of other period weaponry (Tzouriadis and Deacon 2020). Flails of various types were also popular in Switzerland during the late medieval period (Carey et al. 2006). A few select examples of these weaponised threshing flails are presented for context.

Folio 25v from the early 15th century manuscript Besançon BM MS.1360 Bellifortis, by Konrad Kesyer (Germany, Figure 8) illustrates a weaponised threshing flail alongside a halberd or bardiche-type axe being wielded by two plate-armoured knights riding in a war wagon, with a more conventional war flail in the hands of the rider of the horse pulling the wagon in a single image.

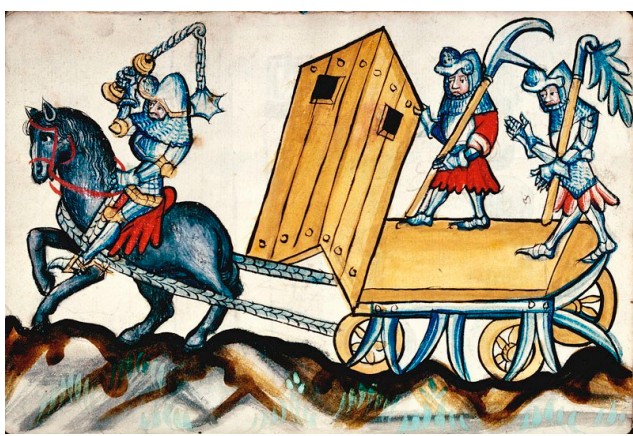

**Figure 8.** Fol. 25v from Besançon BM MS.1360 Bellifortis. (© Besançon Bibliotheque Munucipale, MS 1360, permission granted for non-commercial reproduction, image from https://bvmm.irht.cnrs.fr/iiif/7822/canvas/canvas-1256503/view, accessed on 25 October 2023).

The former and the halberd are wielded by men riding a war wagon, possibly of Hussite origin, as these types of flail were popular amongst the Hussites during their prolonged struggle for independence (Ławrynowicz and Nowakowski 2008; Beňa 2014; Górski and Wilczyńska 2012), along with a plethora of other weaponry (Antoche 2004). An account is provided describing the famous use of the flail by the Hussites as part of their "Wagenburg" (wagon fortress)—each wagon had a crew of twenty: six crossbowmen, two culverins (handguns?), two pavoisiers (shieldmen?), four men with flails, and four with polearms (halberds, etc.) (Biederman 2014; Moreno 2015; Górski and Wilczyńska 2012). Several wagons would then form a circle to create the Wagenburg, from which the Hussites would fight as a makeshift fortress (McLachlan 2011; Górski and Wilczyńska 2012). In this setting, the flails are believed to have been primarily defensive in nature, and given the inability to use a shield, some degree of body armour was also worn by their wielders (Grabarczyk 2000). Women may also have been part of the crews (Moreno 2015). Skulls found in the modern-day Czech Republic and Slovakia with depressed skull fractures verify the effectiveness of flails on the battlefield (Warner 1968)—as Osgood describes it (Osgood 2005):

> "A flail such as this in the hands of country peasants, who were accustomed to using it, must have been a terrifying weapon which could bash the finest helmets of the Crusaders (against the Hussites) to smithereens).

Górski and Wilczyńska similarly describe a Moravian noble's fear in his words of warning to the King (Holy Roman Emperor), "I am very afraid of peasant's flails" (Górski and Wilczyńska 2012). There are several contemporary and later manuscripts which depict the Hussites with their weaponised threshing flails, notably the Jena Codex (c. late 15th—early 16th century, currently held in the Library of the National Museum of the Czech Republic; see fol. 76r).

Folios from the aforementioned German fight manuals shed some light as to how threshing flails and weaponised threshing flails were employed in a battlefield setting in previous centuries, even though the figures drawn are unarmoured in the majority of these works. A selection of folios from the work of Hector Mair (Opus Amplissimum de Arte Athletica (Cod.icon. 393), Bayerische Staatsbibliothek München, Munich, Germany, 16th century) are presented in Figure 9. Talhoffer's c. 1450 Fechtbuch (MS 78.A.15, Stiftung Preußischer Kulturbesitz, Berlin, Germany; see https://wiktenauer.com/wiki/Talhoffer_Fechtbuch_(MS_78.A.15), accessed on 25 October 2023) contains two folios depicting flail combat but little extra beyond this. A comprehensive assessment of the practical combat use of flails described in these sources is beyond the scope of this work.

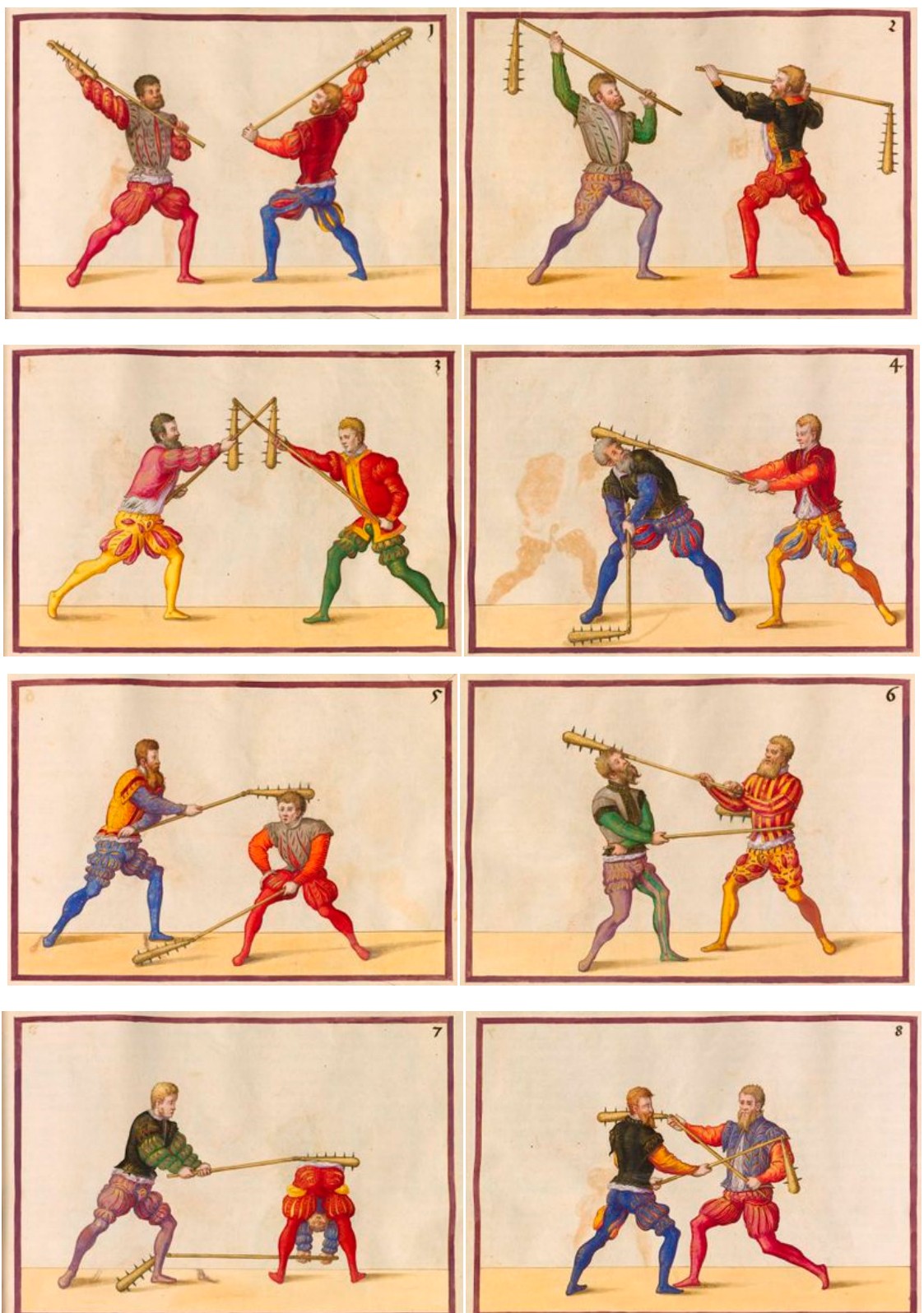

**Figure 9.** Fols. 209r (**1**), 209v (**2**), 210r (**3**), 210v (**4**), 211r (**5**), 211v (**6**), 212r (**7**), and 212v (**8**) from Opus Amplissimum de Arte Athletica. (©Bayerische Staatsbibliothek München, Cod.icon. 393, permission granted for non-commercial reproduction, images from https://wiktenauer.com/wiki/ Opus_Amplissimum_de_Arte_Athletica_(Cod.icon._393), accessed on 25 October 2023).

Interestingly, Talhoffer also depicts what has been referred to as "divorce by combat"—a means of resolving marital challenges by duelling (see Figure 10). (Moreno 2015) As a handicap to the physically stronger man, he was forced to fight with a club-like implement from a 3-foot-wide hole with one hand tied behind his back, while his female opponent had free reign and attacked him with "a rock wrapped in cloth"—in essence a primitive flail (see https://medium.com/lessons-from-history/divorce-by-combat-d9309701f718, accessed on 25 October 23 and https://www.ancient-origins.net/history-ancient-traditions/divorce-combat-0017263, accessed on 25 October 2023). These practices are believed to predate Talhoffer (15th century) by at least several hundred years, but may illustrate the practical implementation of "war flails", albeit in a non-military combat setting. The images presented are from MS. Cod.icon.394a (Bayerische Staatsbibliothek München, Munich, Germany), though additional depictions of this concept are also presented in MS Thott.290.2° (Kongelige Bibliotek, Copenhagen, Denmark).

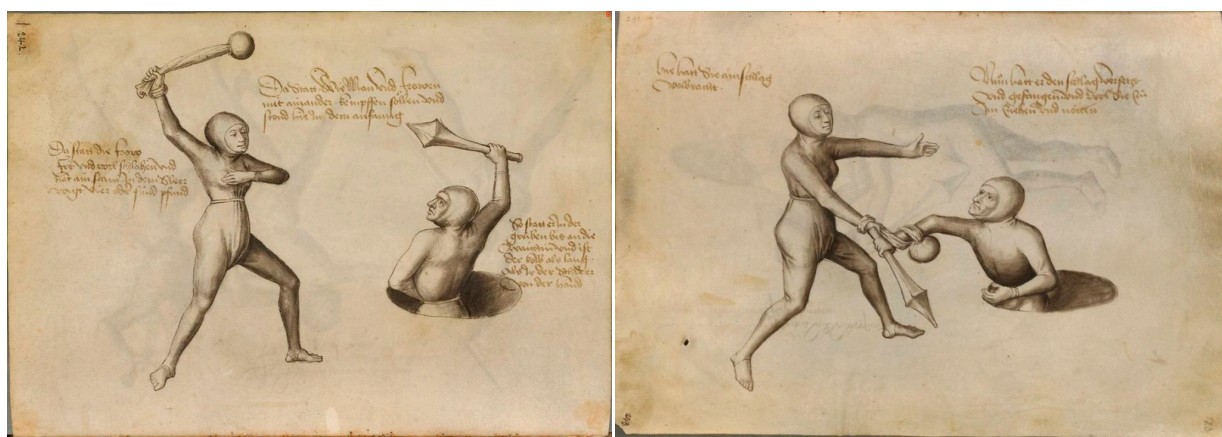

**Figure 10.** Fols. 122v (**left**) and 123r (**right**) from Talhoffer Fechtbuch (c. 1467) illustrating "divorce by combat" with "rock in cloth sack" pseudo-flail. (© Bayerische Staatsbibliothek München, MS. Cod.icon. 394a, permission granted for non-commercial reproduction, images from https://wiktenauer.com/wiki/Talhoffer_Fechtbuch_(Cod.icon._394a), accessed on 25 October 2023).

### 6. A Few Comments on Scourges

At this point it is worth discussing the common confusion between scourges and flails which is quite prevalent in secondary sources (Carver 1986; Hamburger 1991; Baxter 1987; Hahn 1991).

This probably arises as the overarching Latin term *Flagellum* can mean either flail or scourge as a noun or adjective depending upon the context of use, though there are several more specific terms available (*lorum* (whip), *habena* (strap), *scutia* (lash), *stimulus* (goad), *fustis* (staff), *virga* (rod), and *catenae* (chains) (Nicolotti 2017). The similarity in depiction between certain scourges and certain flail designs may also contribute to this (Carver 1986), though it would seem this oft comes down to variability in terminology used (Hamburger 1991), or misunderstandings around the period designs of weaponry, particularly for secondary sources dating from the Victorian era (Waterschoot 2014).

Scourges are, rather than weapons, tools or punishment devices more akin to a whip, or the more modern "cat-o-nine-tails". Scourges are depicted consisting of a wooden handle to which is connected a series of strands, likely leather, rope, or more rarely metal (The History Blog 2016), in which are tied a series of knots (L'Engle 2002) or perhaps bones.

The most common depictions of scourges are not as instruments of war, but rather in a biblical context showing the punishment or chastisement of individuals or creatures (Eberhart 2005); use was also found for self-flagellation in monastic settings from the 1300s (Nicolotti 2017). Several war flails may be misconstrued as scourges and vice versa (see folios from *Arthurian Romances* and the *Life of Eustace and Other Saints*).

Although there are a great many folios, carvings, and statues depicting scourges, most of these are not presented within a warfare or battlefield context, and as such are beyond the scope of this work. Where weapons with some ambiguity between flail and scourge are depicted, these are discussed under the context of war flails, rather than here.

One such example of a depiction that is likely a (three-headed) scourge rather than a flail is fol. 79v from the 12th century English manuscript *Quaestiones hebraicae in Genesim*, etc., also known as *Hieronymi Questiones In Genesim*, etc. (Figure 11). This depicts a naked male armed with a scourge or flail beckoning to a rabbit or hare playing a harp as part of a stylised, large letter "A" (Thompson 2000).

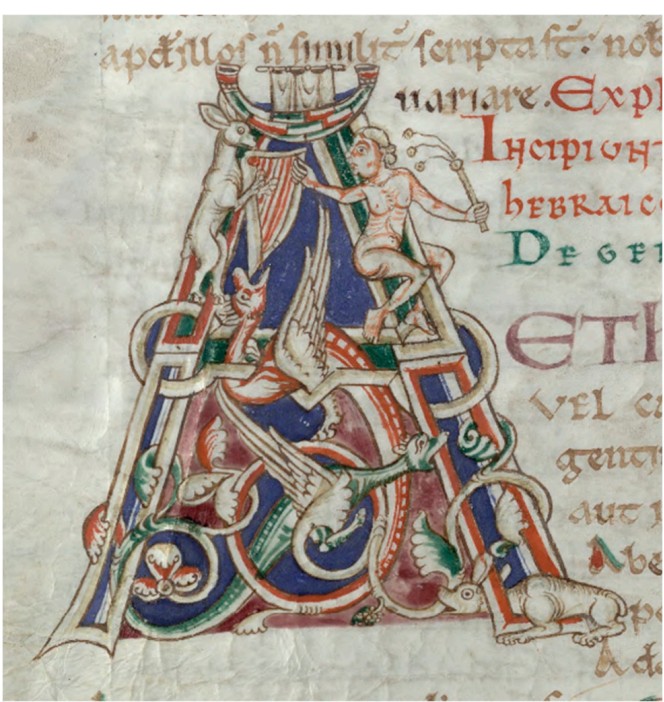

**Figure 11.** Likely scourge or possible flail depicted in fol 79v of *Quaestiones hebraicae in Genesim*. (© The Master and Fellows of Trinity College, Cambridge, Trinity College Library MS B.2.34, permission granted for non-commercial reproduction, image from https://mss-cat.trin.cam.ac.uk/Manuscript/B.2.34/UV#?c=0&m=0&s=0&cv=81&r=0&xywh=704%2C807%2C1403%2C2167, accessed on 25 October 2023).

Fol. 369v from the manuscript BNF Français 152 Bible Historiale—Part 1 (mid-14th century, France, Figure 12) depicts a siege in which two soldiers threaten a woman in their castle (presumed a prisoner) with what appear to be scourges.

One example of the confusion between flail and scourge arises in the early 11th century manuscript known as the *Harley Psalter* (Harley MS 603), in which two folios (f14v and f53) present two different types of what are described as "flails" by Carver (Carver 1986), but probably more accurately represent scourges, especially in the case of the former, as demonstrated by a similarly designed (ornate and likely ceremonial) late-Saxon scourge (Wilson and Blunt 1961), which would be totally impractical as a battlefield weapon, but rather serves as a symbolic religious artefact.

Period illustrators were by no means exempt from this confusion either; several examples exist of different drawings within the same manuscript produced by different artists whose styles differ sufficiently for items depicted (flails and scourges) to be sufficiently distinct (Morey 1931). Similar extant items have also been found and described (The History Blog 2016).

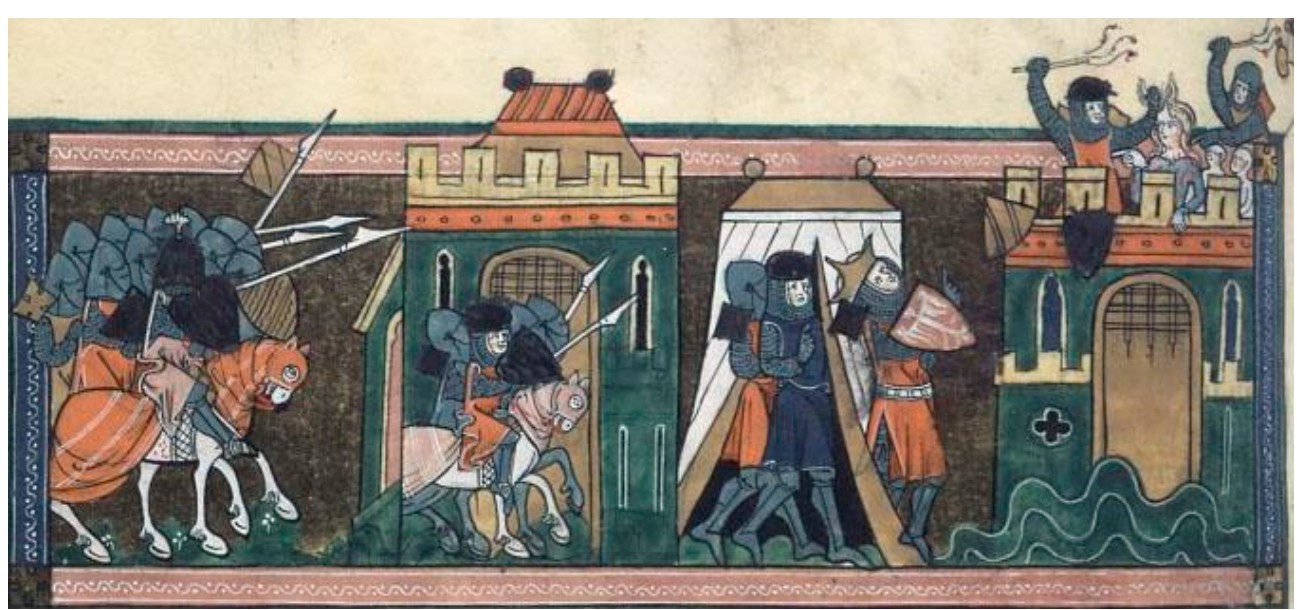

**Figure 12.** Fol. 369v from Bible Historiale—Part I (© Bibliothèque nationale de France, BnF MS 152, permission granted for non-commercial reproduction, image from https://manuscriptminiatures. com/4246/8208, accessed on 25 October 2023).

## 7. Development and Use of War Flails Outside of Western, Central, and Southern Europe up to the Early and High Medieval Eras

It is pertinent before we analyse the bulk of European sources covering war flails to discuss the wider context of these weapons outside of this region of interest, as this likely had a significant impact on the development and adoption of these weapons within Europe.

War flails (known commonly as *kistens*—a word with either a Slavic or Turkic root (Kotowicz 2006))—were prevalent between the 5th and 10th centuries in the Asian Steppe (Bulgars, Avars, Khazars, Altai, etc.), primarily as a secondary weapon for cavalrymen who, during their contact with the Kyivan Rus and other Eastern European peoples, adopted these weapons during the 8th through 11th centuries, when their use by infantry also expanded (Taavistainen 2004; Shpakovsky and Nicolle 2013; Zhirohov and Nicolle 2019; Kotowicz 2006, 2008).

The cultural contact between West and East in the 11th to 13th centuries meant that certain Western technologies and concepts were passed Eastwards while Eastern ones, such as flails, were passed Westwards (Kirpičnikov 1968). Use of flails in Central Europe (Poland primarily, but also parts of the former Holy Roman Empire—modern-day Germany) was prevalent from the 10th to the mid-14th centuries (Puziuk and Tyniec 2013), when "Westernisation" of arms and armour became ever-more common (this development is discussed later) (Kotowicz and Skowroński 2020; Kotowicz 2008), though use in modern-day Russia and Ukraine (Novgorod, etc.) continued well beyond this (Drob and Vasilache 2022). Such weapons were not just imported into Central Europe from the East—local production of flails was common at scale during the 11th and 12th centuries (Kotowicz and Michalak 2007; Michalak 2006), though Polish nobles were known to import weapons, including ornately decorated flail heads (Kotowicz 2006). The designs of flail heads across Eastern Europe and the Western Asian steppe evolved over time as contact with new cultures and armour technologies demanded it, with "feedback" occurring in the styles observed reflecting Eastwards migration of newer trends in materials (Sitdikov et al. 2015); the patterns of flails adopted in Eastern Europe did evolve as the technology became entrenched after being adopted from the nomadic peoples that originated them (Kirpičnikov 1968). The Mongols of the Golden Horde were also known to use flails (Culic 2016), which may have more closely resembled weaponised threshing flails in design, with elongated

metallic heads rather than wooden ones (Kuleshov 2019), similar to those perhaps used in antiquity by the Thessalians.

Eastern European flail heads come in a wide range of designs and degree of decoration, and were initially made of bone, horn, antler, or stone, or perhaps wood, up to the 11th century (Rabovyanov 2021; Kotowicz and Michalak 2007; Grotowski 2010; Michalak and Zamelska-Monczak 2016), from which construction began to transition to metal (brass, bronze, iron, and lead) (Taavistainen 2004; Shpakovsky and Nicolle 2013; Kotowicz 2008). Composite heads consisting of organic material (bone/horn) with an iron insert have also been found and described (Zhirohov and Nicolle 2019). *Kisten* heads are either constructed with a central hole or an attached loop through which the flexible portion of the weapon (rope, chain, or leather thonging) can be attached, but given the perishable nature of organic materials, these are rarely found with their associated heads, as these components have decayed with centuries of exposure to the elements (Kotowicz 2008; Michalak and Wolanin 2008). In Eastern European use, *kistens* typically had a 40–50 cm wooden handle, to which was attached the flexible component of the weapon and ultimately the head were attached, although the thong or rope could be looped around the hand alone in lieu of this (Kotowicz 2006). The early organic heads were likely carved, whereas the later metallic ones were either forged or cast depending upon the metal used (Zdaniewicz and Adamiak 2011).

Despite the relative lack of illustrations or textual sources referring to *kistens* in Eastern Europe (compared to the rest of the continent), hundreds of physical finds have been discovered dating from the 10th to 13th centuries, extending from modern-day Russia and Ukraine into the Baltic states, the Balkans (as far West as Slovenia and Croatia) (Culic 2016; Rabovyanov 2021; Grotowski 2010; Popov and Aladjov 2016; Tihle 2017; Svetec 2023), Finland, and Poland (Kotowicz and Michalak 2007), where direct trade and contact occurred with the Rus and the Norse, given the fluidity and porosity of medieval borders (Taavistainen 2004). The extent of flail head finds in Eastern Europe has been extensively documented, particularly in more classical literature dating from the Soviet period (Kirpičnikov 1966), with maps of the finds produced and catalogued (Osypenko 2019), and typologies of the style of head developed, though these do have their limitations (Kotowicz 2008; Osypenko 2019, 2020), the discussion of which is beyond the scope of this work. Please see the many excellent Eastern European references for a summary of these.

Flails were also prevalent in the Middle East in the early medieval era, finding use in pre-Islamic Iran (6th–7th century and before) (Phillip 2019), while also being adopted as an anti-armour weapon by the Turks (as a secondary alongside the bow) (Zouache 2007; Phillip 2019). The works of the Islamic scholar and historian Ibn al-Athir (1160–1233) attest to a flail being carried and used to great effect by the Seljuk Sultan at the Battle of Manzikert (1071) (Alatas 2018).

## 8. Depictions, Descriptions, and Archaeological Finds of War Flails in Western, Central, and Southern Europe in the Early and High Medieval Eras

In contrast to Eastern Europe and the Balkans, where physical finds of flail heads predominate, in Western, Central, and Southern Europe, primary textual and artistic references (carvings, statues, illuminated manuscripts, etc. (Figure A1)) are far more common (Hrynchyshyn 2014; Taavistainen 2004; Kotowicz and Skowroński 2020; Imiolczyk and Zdaniewicz 2022; Shpakovsky and Nicolle 2013). The reasons for this may be twofold: firstly, the challenges arising in accessing Eastern European textual and artistic sources due to language barriers and present geopolitical tensions; and secondly, the misidentification of many archaeological finds outside of the Eastern regions of Europe, which may in fact be flail heads rather than weights or other artefacts, as discussed later. This does not exclude confusion of such archaeological finds in Eastern Europe, however (Florek 2019).

Soler del Campo notes that war was an integral part of medieval life, with weaponry as such constituting a significant part of the material, cultural, and technological facets of the time, serving as essential tools for survival and status symbols both (Soler del Campo 1985). The study of weapons, in this case flails, and their evolution over time, must thus

take into account societal and cultural factors in addition to the technological and tactical ones (Soler del Campo 1985). These kind of studies require a three-pronged approach investigating archaeological, iconographic or artistic, and historiological sources to grant a comprehensive understanding (Soler del Campo 1985). Using this methodology, I have attempted to present and analyse the numerous and varied primary (and some secondary) sources which depict and describe flails which are available to me. While I have attempted to highlight general trends in flail design evolution over time, a thorough typology of these weapons is beyond the scope of this work. Readers will note the somewhat extensive literary references to flails in the various medieval national matters, for to quote Jean Bodel (d. c. 1210) in *La Chanson des Saisnes* (Song of the Saxons, taken from the translation by Stengel and Menzel (Stengel and Menzel 1906)):

> "Ne sont que III matieres a nul home antandant: De France et de Bretaigne et de Rome la grant."

> "There are only three subjects matters for any discerning man: That of France, that of Britain and that of great Rome."

I will highlight references to flails where present within the various *chansons*, poems, and other works of folklore, but please note that this is not an exhaustive list but based more upon the abundant secondary sources which cite and mention these; translations are approximate based upon the availability of original texts, translated versions, and dictionaries, where these are available. Furthermore, some of the poetic or textual references may predate their first recorded written examples by a significant amount of time (decades or even centuries) when drawn from oral histories of their parent peoples, as these were commonly adapted and changed over time. Where possible, a commentary of the apparent construction of the weapons depicted and on the reliability of the source(s) is provided. The sources are presented in approximate chronological order. In some cases, the original texts are not available, where I would refer readers to the cited secondary sources for guidance. Several questioned depictions are also highlighted, and where these have been suspected over the years, this is discussed along with previous assessments as to their authenticity.

Flails are described in use by the Bretons to fight off the Northmen (Viking raiders) during the late 9th and early 10th century in a ballad referring to the Breton leader Alain Barbe-torte (Alan II, Duke of Brittany, c.900–952). Flails are described thusly (in the original full Breton text, translated in whole into French (Villemarqué 1846), and from that to English). The type of flail utilised is not comprehensively described, but the reference to iron suggests this is a weapon rather than a threshing flail as described in some other folk tales later. The "Saxons" mentioned here are the Norse, due to the related (Germanic family) languages spoken by the two peoples (Spence 1917).

> "Ar Vretoned a weliz o vac'h el leur e louc'h
>
> Ken a lame pellenou demeuz ar pennou blouc'h
>
> Ha ne ket gant fustlou prenn a vac'h ar Vretoned
>
> Nemet gand sparrou houarned ha gand tried ar virc'hed."

> "J'ai vu les Breston batter le blé dans l'aire foulée
>
> J'ai vu voler la balle arrachée aux épis sans barbe.
>
> Et ce n'est point avec des fléaux de bois que batten les Bretons
>
> Mais avec des é[ieux ferres et avex les pieds des chevaux.

> "I saw the Breton batter the wheat in the stomped area
>
> I saw the chaff fly off the beardless ears
>
> And it's not with wooden flails that the Bretons strike
>
> But with iron spikes and horses' feet."

Flails (reported as schlachtgeissel—lit: "battle whip") are attested as being used in 10th century Germany during the second Battle of Lechfeld (955) in which the Kingdoms of

Germany and Hungary clashed; these are described as consisting of a short wooden handle with three or four chains, at the end of which an iron ball studded or filled with lead was attached (Köstler 1883; Bánlaky 1928). This account may arise from the *Annales Altahenses* or similar period document (Pohl 2004; Négyesi 2003). Bánlaky suspected that the *Schlachtgeissel* originated from the Frankish Empire, and its use was passed onto the Carolingian successor states, used as a secondary or tertiary weapon by the German cavalry (Bánlaky 1928; Jócsik 2022), as was also likely the case for the steppe peoples who first introduced such weapons (*kistens*) to parts of Central Europe (Zhirohov and Nicolle 2019). The flail may have been introduced to the Carolingians by the Huns (Gotzinger 1885; Jähns 1880). If this is the case, a likely explanation is provided for the significant prevalence of flails in the many tales of the Matter of France, England, and further afield. A plethora of 10th and 11th century grave finds from present-day Central and Eastern Hungary would support the notion that flails were also used by the Magyars (fighting for or with the Hungarians) during this time period (Jócsik 2022).

The Welsh poem *Geraint uab Erbin* (Geraint, son of Erbin) is believed to originate from the 10th or 11th century, but the earliest surviving version dates from c. 1250 (the Black Book of Carmarthen, in Welsh) (Bollard 1994). A dwarf character in the poem is described as using a flail (type not specified) to "whip the queen's companion" (a knight) (Fee 2018). I do not have access to a suitable translation of original text (available http://www.maryjones.us/ctexts/bbc22w.html, accessed on 25 October 2023); this reference is otherwise taken from the work of (Fee 2018).

Courville claims that the "morning star" style flail was introduced into England with the Norman conquest and was used for four hundred years following this, alongside extensive adoption in Teutonic (German) and Slavic Europe (Courville 1948). For the first two cases, the limited evidence of flail-type weapons (known examples and references presented here) render his position for the first two points obsolete—we can verify a limited knowledge and adoption of flails in England and Western Europe, but not to the significant extent observed in Slavic lands. Hewitt considers that "stone flails" were used by the Saxons at Hastings, but this is likely a misinterpretation of the source (William of Poitiers, d. 1090) cited (Hewitt 1855). This demonstrates the inaccuracies sometimes incurred in dealing with secondary sources, especially those from Victorian times and less recent works, in addition to those where the authors are not subject matter experts regarding their writings (Hewitt 1855; Cuming 1854; Cowper 1906; Demmin 1911; Snook 1979), though there are outliers in this respect (Schultz 1889; Viollet-le-Duc 1874; Sternberg 1886) who demonstrate an excellent comprehension of historical context and nuance. Modern advances in science and understanding around medicine and historical recreation and interpretation have further allowed us to cast doubt on certain more classical sources in light of new evidence and lack of a "rose-tinted view" of the medieval era (Geldof 2015).

A war flail is reportedly depicted in a carving (statue) of one of the founders (*Stifterfiguren*) of Naumburg Cathedral (Germany, Figure 13), possibly dated to the late 11th century (Moreno 2015), though this may be mis-dated and actually be a later creation from the 12th or early 13th centuries, or fabricated in secondary sources of dubious accuracy (Dona 2021; Schultz 1889; Demmin 1911). The weapon depicted is a ball-and-chain type flail with a smooth ball. I have been unable to find more recent images of this purported depiction. The other items depicted in the figure do provide a spread of late medieval flail designs, however, some of which are claimed to be based on extant objects.

One of the earliest depictions in Southern Europe is a late 11th or possibly early 12th century mosaic depicting the "Kiss of Judas" or the "Betrayal of Jesus" from the Basilica di San Marco (St Mark's Basilica, Venice, Italy, Figure 14), depicting a mob armed with an assortment of weaponry including a ball-and-chain flail, a spiked flail, and a very uncommon double-headed flail alongside lanterns, torches, and two-headed axes. The colours present in the artwork suggest the author intended to convey the use of metal (iron) flail heads on wooden hafts. As these are in the background of the image, no more detailed information regarding the dimensions of these weapons can be ascertained. This does

demonstrate knowledge of these types of weapons in Italy by the end of the 11th century at the earliest, or the early 12th century at the latest. Their presence in a mob-like setting reinforces the stereotype that such weapons were used by pagans or heathens (Tihle 2017).

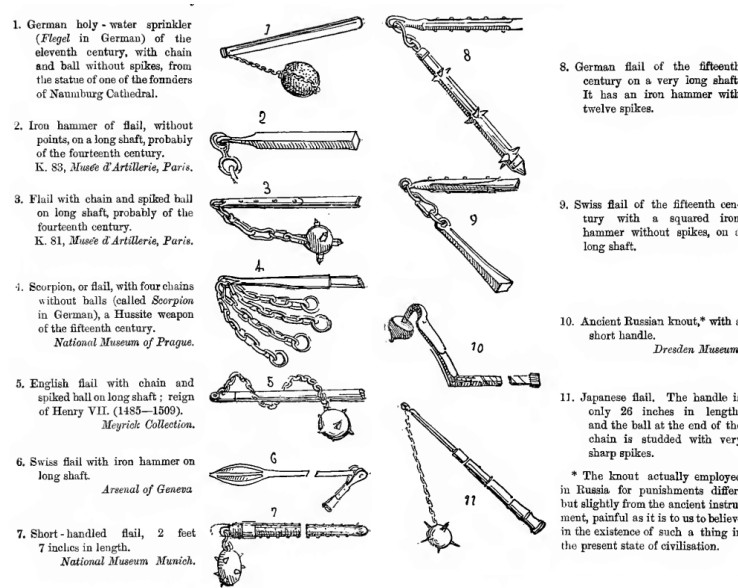

**Figure 13.** Depiction of the flail depicted in Naumburg cathedral (top left). A selection of later period war flails are also presented in this image. This figure originates from one of several 19th century works in French and German, and has been copied extensively. (Image from Campagnano 2015— https://zweilawyer.com/2015/05/31/armi-immanicate-da-botta-ii-il-mazzafrusto/, accessed on 25 October 2023).

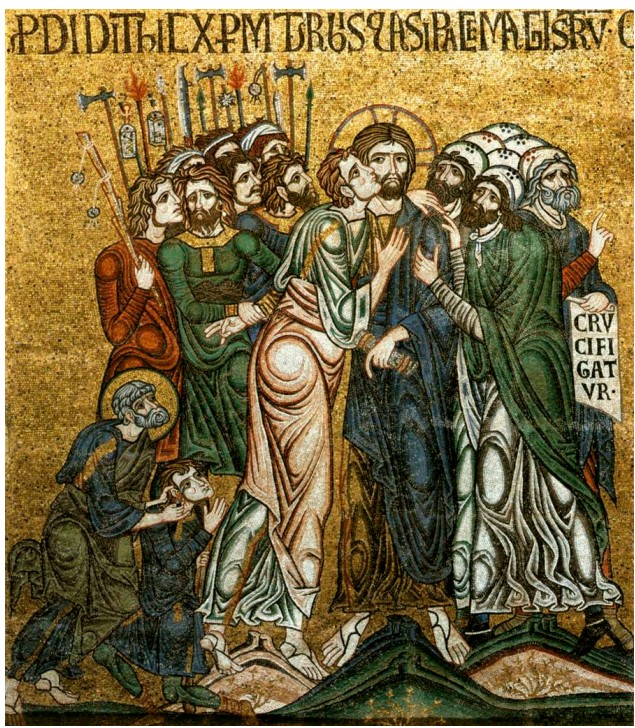

**Figure 14.** Mosaic from the Barrel Vault, Basilica di San Marco, Venice. (Image provided courtesy of the Web Gallery of Art (http://www.wga.hu/, accessed on 25 October 2023), specific image from https://www.wga.hu/art/zgothic/mosaics/6sanmarc/3barrel5.jpg, accessed on 25 October 2023).

A similar depiction is presented in the Queen Melisende Psalter (fol. 7v in BL Egerton MS 1139, Figure 15), believed to have been prepared in Jerusalem c. 1130–1140, as in part of a mob arresting Jesus, one of the weapons depicted is clearly a ball-and-chain flail, in addition to spears, maces, and torches. The colouring of the folio indicates the flail depicted has an iron head and chain on a wooden haft.

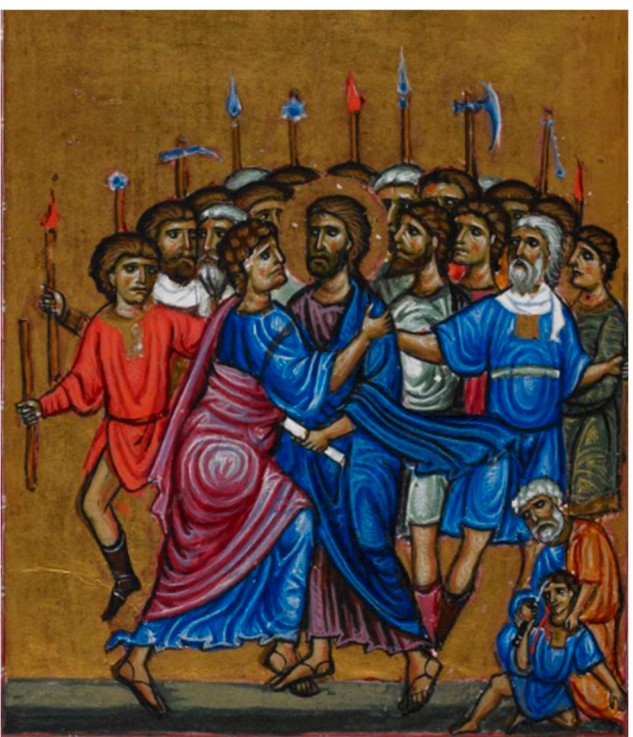

**Figure 15.** Fol. 7v from the Queen Melisende Psalter (© British Library Board, BL Egerton MS. 1139, permission granted for non-commercial reproduction, image from https://www.bl.uk/manuscripts/Viewer.aspx?ref=egerton_ms_1139_fs001r, accessed on 25 October 2023).

War flails are not just depicted in military or mob-like settings; several images of demons wielding flails are present in both illuminated manuscripts and carvings present on churches. For example, L'Eglise Sainte-Foy de Conques (Abbey Church of Sainte-Foy, Conques, France, Figure 16), completed c. 1125, contains an elaborate carved arch above the entranceway, depicting a battle between angels and demons, the latter of which are armed, amongst other things, with a long shield, an axe, and a war flail (Figure 17) (Deschamps 1941; Denny 1984; Moreno 2015). This depicts a scene entitled "The Last Judgement", as per the depiction in the *Winchester Psalter* subsequently described.

Similar depictions are shown in the *Winchester Psalter* (c. 1150, England, BL Cotton MS Nero C IV, Figure 18), where, during a scene titled *The Final Judgment*, a pair of demons armed (centre and left of the figure) with war flails are presented tormenting the damned alongside several others armed with pitchforks (Denny 1984; Moreno 2015). The form of the flails presented here is the ball-and-chain weapon type, rather than scourges (discussed earlier) as are oft-depicted in monastic or biblically focused manuscripts; the handles appear to be ~2/3 the length of the overall weapon, with the flexible portion being the remaining 1/3 of the total length. This demonstrates a certain awareness of these devices within England by the mid-12th century. Collectively, these sources alongside those depicting the Kiss of Judas paint flails as an instruments used by the unholy, which represents a majority opinion until around 1250, as discussed later.

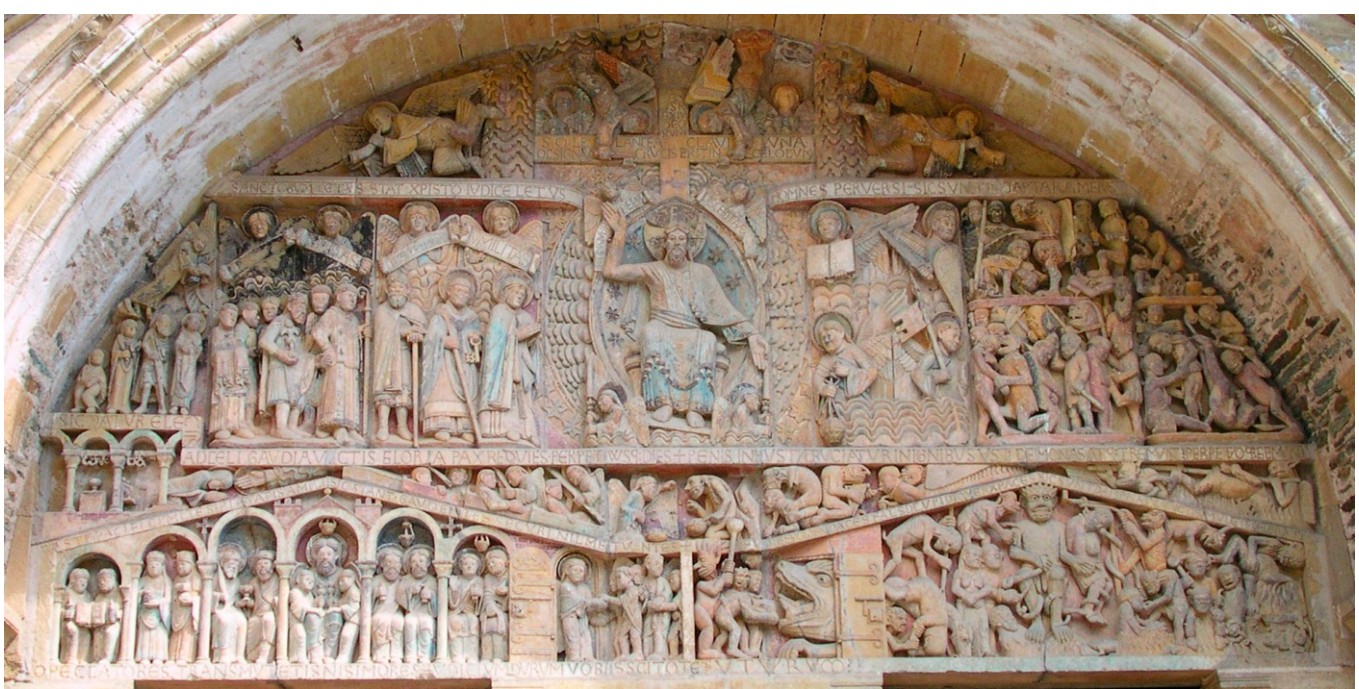

**Figure 16.** Carving from Sainte-Foy Abbey entranceway. The demon with the flail appears in the upper-right hand side of the doorway carving. (© Jean-Louis Zimmerman, image sourced from https://www.flickr.com/photos/jeanlouis_zimmermann/225026827/in/photostream/, accessed on 25 October 2023 under CC BY 2.0 license for non-commercial reproduction).

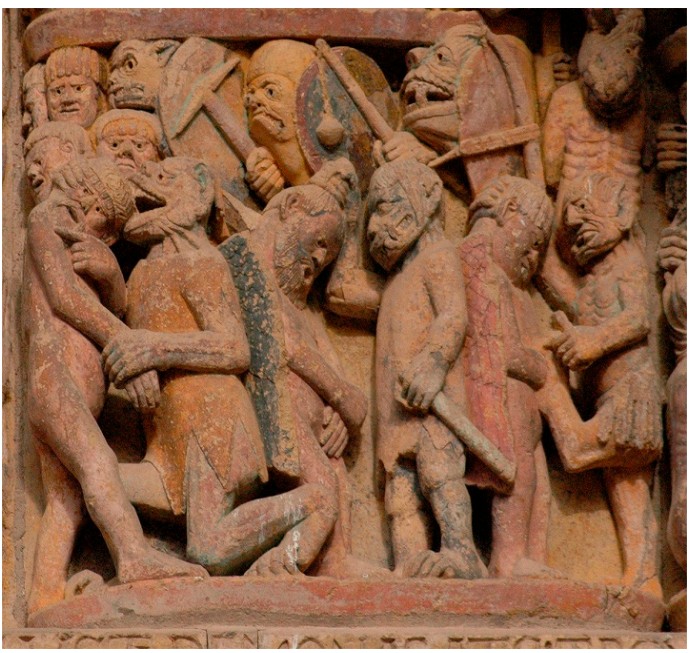

**Figure 17.** High-resolution image of carving of demon with flail from Sainte-Foy Abbey entranceway. (© www.monestirs.cat, accessed on 25 October 2023, image kindly provided with permission for non-commercial reproduction).

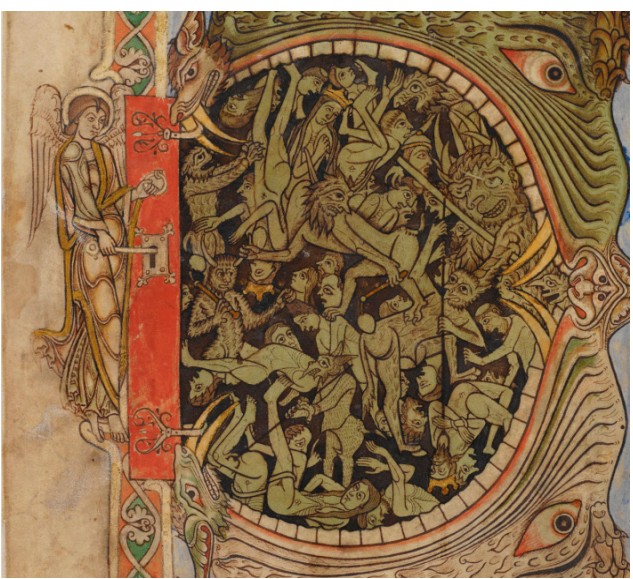

**Figure 18.** Fol. 39r from the *Winchester Psalter* (© British Library Board, BL Cotton MS Nero C IV, permission granted for non-commercial reproduction, image from http://www.bl.uk/manuscripts/ Viewer.aspx?ref=cotton_ms_nero_c_iv_f039r, accessed on 25 October 2023) (Denny 1984).

The Anglo-Norman *Roman de Brut* (c. 1155), translated and adapted from the Latin *De gestis Britonum or Historia Regum Britanniae* (History of the Kings of Britain) by Geoffrey of Monmouth (c.1095–1155) by the poet Wace (c.1110–1175) describes *plomées* being used during a siege, (Le Roux de Lincy 1836) though in this context, this likely refers to leaded projectiles possibly hurled from slings, rather than flails, once again demonstrating the multifaceted meanings of many medieval words and the confusion this can cause when analysing such texts. This is drawn from a complete translation by de Lincy (Le Roux de Lincy 1836).

"Li Romain as murs les atendent

Qui à mervelle se desfendent. . . (3090)

. . .Lancent dars et plomées ruent

Maint en abatent et maint tuent"

"Romain had walls waiting for them

Which were wonderfully defensible. . .

. . .Arrows flew and [lead (ed projectiles)] were thrown

Many were cut down and killed. . ."

The 12th century manuscript *La Chanson de Jerusalem* describes flails (possibly threshing flails (*plomées*), though these are also described as "large clubs with dangling chain") being wielded by the Tafurs (Frankish Catholic zealots who adopted a vow of extreme poverty) during the 1099 Siege of Jerusalem. The Saracens are described as being similarly armed (perhaps with ball-and-chain flails), one of whom "smashed a flail down on the King of the Tafurs. . . (leaving him) dripping with blood, the blow from the flail having smashed his nose and badly damaged his head and brains". Two Christian knights (Eurvin of Creel and Wicher the German) are noted as being similarly set upon by "infidel pagans. . . wielding large thick maces and hinged flails, big lead clubs. . ." This piece clearly describes flails of various types being used by both sides during the First Crusade (1096–1099) (Sweetenham 2016; Esposito 2018). The following excerpts are drawn from the presentation of Esposito (Esposito 2018).

"Ha! Dex! La n'ot mestier gius ne gabbi ne ris.

Bauduïns de Belvais fu navrés ens el pis

Et Harpins de Boorges devant en mi le vis

Et Ricars de Calmont estoit el cief malmis

Et d'une grant plomee ferus Jehans d'Alis

Si que li ber en ert encor tos estordis. (vv. 2374–2379)."

"Anuit m'est avenus uns damages mortals,

Al besoing m'ont fali nostre malvais deu fals.

Mais tant les ferai batre de fus et de tinals

Et de maces plomees, de bastons et de paus

Que ja mais n'aront cure de tresces ne de bals! (vv. 1762–1766)"

"Cascuns porte en se main u maçue u baston,

Plomee u materas et piçois u bordon

U gisarme aceree u grant hace u piçon. (vv. 1823–1825)"

"Es vos le roi tafur par mi .I. sablonal

A .X.M. ribaus: cascuns tient hoe u pal

U gissarme u picçois d'acier poitevinal.

Portent mals et flaiaus, fondefles et mangal. (vv. 1982–1985)"

"Portent haues et peles et grans fausars et pis,

Gisarmes et maçües et mals de fer traitis,

Trençans misericordes et cotels couleïs

Et plomees de coivre a caaines assis.

Li auquant portent fondes, molto nt caillos coillis (vv. 3012–3016)"

"Es vos le roi tafur et dant Pieron corant

Et Tafurs et Ribals qui molt vienent huant:

N'i a cel ne port hace u macüe pesant,

Coutel u grant plomee a caaine pendant,

U pouçon u piçois u alesne poignant.

Li rois tafurs tenoit une grant fauc trençant,

Entre paiens e mist, tant en vait craventant

Que par mi les ocis ne pot aler avant. (vv. 5912–5919)"

"Des mors et des navrés vont la terre covrant,

Fors de Jerusalem les mainent reculant.

Tos tant fierent sor els a tas demaintenant,

As grans maces de fer les vont jus craventant

Et as grandes plomees contre terre tuant.

Del sanc as Sarrasins i ot plenté si grant

Contreval le fossé en vont li riu corant. (vv. 7546–7557)"

The *Song of Roland* (Chanson de Roland), is a French epic poem, possibly written by the poet Turold (Turoldus in the manuscripts themselves) around the turn of the 11th century (est. 1040–1115), which describes the battle of Roncevaux (778) from the time of Charlemgane (Thibout 1966). One of the characters in the poem, Olivier (Oliver), is described as using a

ball-and-chain flail, with depictions of this character in various medieval locations including as a gate guardian at the Cathedral of Verona, Italy (c. 1139, Figure 19) (Monfrin 1965; Thibout 1966; Beaud 2017; Spiro 2014; Moreno 2015; Agrigoroaei 2018), depicted opposite Roland, who is more conventionally armed (with sword and shield) and armoured in maille.

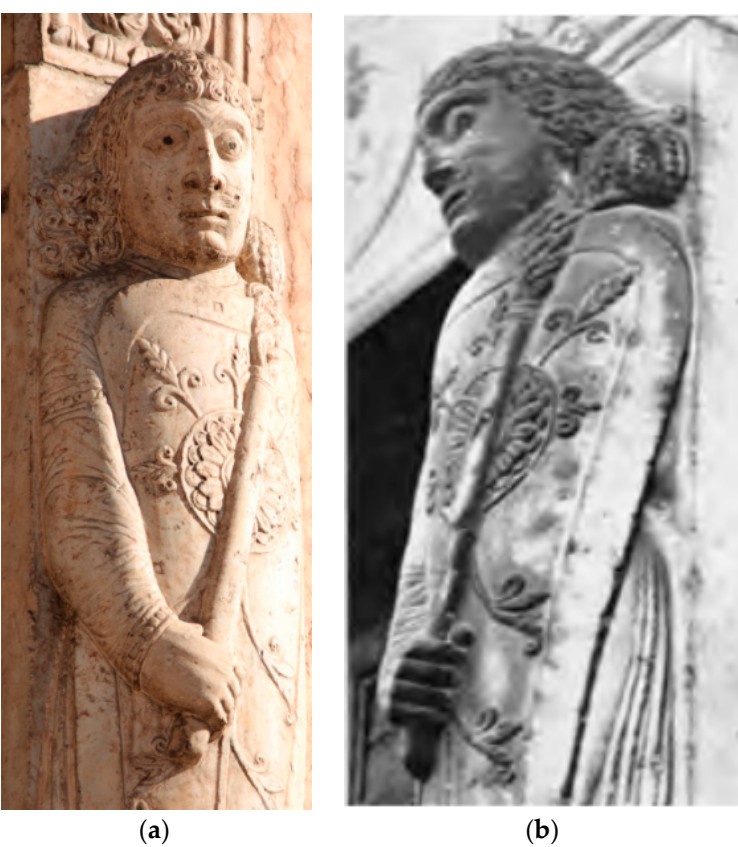

(**a**)                                  (**b**)

**Figure 19.** Carving from the gate guardian statue on the entranceway of the Cathedral of Verona, depicting Olivier from the *Song of Roland* with a flail, Verona, Italy. (Images from Agrigoroaei 2018 (**a**), and Dona 2021 (**b**); reproduced under CC-BY 4.0 license).

As the *Song of Roland* was composed around the time of the second crusade (1145–1149), coinciding with the liberation of parts of Spain and Portugal from the Moors and thus mirroring the deeds of the champions (or perhaps nephews?) of Charlemagne (Roland and Olivier), the time from which the story is believed to originate (Spiro 2014; Meneghetti 1990). Roland is postulated to have been the nephew of Charlemagne, while Olivier could be an allegory for St. Romanus (Dona 2021). The translated text of *the Song of Roland* does not itself mention the flail (Rabillon 1885). The characters in period poems such as the *Song of Roland* are often transcribed into biblical or religious settings when illustrated or turned into physical carvings, with associated religious meanings attached to them (Kahn 1997). Beaud notes that in much of the epic literature (*c.f.* the Matters of France and Germany, and biblical depictions), flails are most commonly shown being used by "bandits, peasants, and pagans... are pejorative for the wielders..." and demons or devils, as I have previously demonstrated (Beaud 2017). In use by the paladin Olivier, the meaning of the flail may be reversed, perhaps as an iconographic representation of divine rebuke transferred by the blows of the weapon (Beaud 2017). There has been speculation as to why Olivier is depicted unarmoured (Dona 2021), but the fact that he is carved carrying a shield indicates the stance of a warrior, rather than any more pacifistic option. Several archaeological finds present in the Abbey of Rencesvalles (Spain) are attested by some 16th century sources to the flail of Olivier, but these are believed to be late medieval replicas or forgeries (Dona 2021).

The mid-12th century French epic *Prise d'Orange* (lit. "Taking of Orange", originally written in Old French) describes the exploits of the hero Guillaume, who captures the walled city of Orange (France) from the "Saracens" (Moors; see below). Guillaume's wargear is described in somewhat exaggerated terms in the original text, and includes, amongst other things, a *fléau d'armes*, though this is referred to as a *plomée* in the period text (Cazanave 2013; Sternberg 1886). As with many of the similar French chansons, the origin of this particular story dates from the time of Charlemagne. An excerpt from the *Prise D'Orange* (IV, 241) (Schultz 1889) is presented. I am unaware of any readily available versions of the full text or modern translations.

"S'o ot plomées et maint fanssart pesant

Et maintes maces et espées tranchanz"

Some degree of confusion arising from the multiple meanings derived from certain words can be evidenced by references to the *plomée* in the *Roman de Thebes* (Romance of Thebes), a mid-12th century (c. 1150–1155) French epic poem believed to be an abbreviated and adapted version of the classical *Thebiad*. The story is fitted to 12th century French methods of warfare and chivalric code, and includes several references to the use of a *plomée* (Constans 1890; de Lage 1966), at least one of which refers to something akin to a discus (v. 230–234 presented below) (Mosca 2020; Ferlampin-Acher 2007), while others are clearly in reference to the "leaded mace" or flail (v. 33–37 and 4175) (Mora-Lebrun 1995). This story was also likely adapted into the late 12th century Anglo-Norman *romans d'aventures* entitled the *Ipomedon* and the sequel *Protheselaus*. These excerpts are taken from the full-text provided by de Lage (de Lage 1966).

"N'i avoit pas esté grant pose (230)

que es geuz sort une mellee

par la raison d'une plomee (discus)

que li danzel iluec gitoient,

qui de giter mout se prisoient"

"Car il n'ont pas escus de chesne,

Espiés de fer, hanstes [de] fresne,

Elaives ne lances ne espees, (2720)

Maces de fer ne granz plomees (flail)

For solement danz Jupiter,

Qui tint un dart agu de fer."

One of the earliest examples of a multi-headed flail is presented in a folio (97v) from the German manuscript *Stuttgarter Passionale—Par Hiemalis* (cid.bibl.fol.57, Figure 20), dated variously between 1130 and 1175. This folio depicts an unarmoured man wielding a three-headed ball-and-chain flail in one hand. The illustration is lacking the colour of some other depictions, but we can infer from the black hue of the balls and chains that this was intended to represent construction of iron. The balls of the flail present are round in nature, rather than spiked as in some contemporary depictions. The digitised images cut off two of the balls of the flail, but these are clearly visible on the opposite page, as pictured. As with one of the threshing flail depictions presented above, this image is worked into a highly stylised letter.

The use of flails outside of Western, Central, and Southern Europe continued in earnest throughout other parts of the known world, including the Byzantine Empire. Several folios from the *Madrid Skylitzes* (dated mid-11th century) present images of weapons which are likely war flails (D'Amato 2011), some of which include the spiked ball varieties. This is backed up by significant archaeological evidence (flail head finds) from the former territories of that state, especially in the Balkans (Culic 2016; Rabovyanov 2021). The well-characterised cultural contact between the Norse, Rus, and even English with Byzantium,

in the form of the Varangian Guard and similar mercenary formations within the Byzantine Empire, also resulted in foreign technologies being brought into the Empire and retained for centuries (Katona 2017). The prevalent interactions between the Byzantines and the Western and Southern European powers during the 11th, 12th, and 13th centuries mean that a degree of technological and cultural awareness of each other's capabilities and habits must have formed during this period, especially given the tendency of certain peoples (Normans) to serve as mercenaries across much of the Mediterranean.

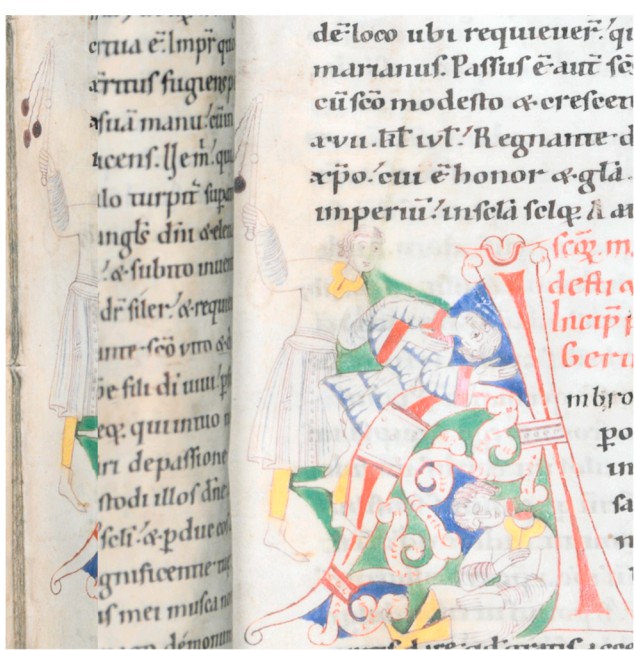

**Figure 20.** Fol. 97v from the *Stuttgarter Passionale*—Par Hielamis. Dual-spread cropped to illustrate three-headed flail. (Württembergische Landesbibliothek, MS. Cod.bibl.fol.57, reproduction under Public Domain 1.0 license, images from https://digital.wlb-stuttgart.de/sammlungen/ sammlungsliste/werksansicht?tx_dlf%5Bdouble%5D=1&tx_dlf%5Bid%5D=11608&tx_dlf% 5Border%5D=title&tx_dlf%5Bpage%5D=206&cHash=2832f875f3ffe62150b0fca282900827, accessed on 25 October 2023).

As discussed above, the Byzantines may have adopted flails from the Khazars, who served as mercenaries in the Imperial Guard for centuries, or adopted from their Bulgar opponents in the 10th and 11th centuries (Grotowski 2010). These flails are occasionally confused with scourges in illustrations but have been described in use as (horseback) battlefield weapons—consisting of a wooden handle to which small (5 cm or less) bronze or antler, stone, or bone balls were attached by leather or hemp rope. A great many depictions of such weapons appear in the *Madrid Skylitzes* (D'Amato 2005, 2011; Rabovyanov 2021), though from the low quality of the images available, these could represent either flails or maces, though they are rather long to be the latter. See the 2011 work of D'Amato for a comprehensive selection of these and https://www.loc.gov/item/2021667859, accessed on 25 October 2023 for the full manuscript. Finds from the 11th century around Romania verify such descriptions (Culic 2016). A Norman mercenary captain is described as being used to beat (humiliated) a Byzantine officer with a flail in several sources from both sides (D'Amato 2005).

The c. 1170 *chanson de geste* entitled *La Bataille Loquifier* (the Battle of Loquifier, adapted from MS de l'Arsenal 6562), which is part of the Cycle of Guillaume d'Orange, includes several references to the *plomée* (flail) within the full text presented by Rubeburg (Runeburg 1913):

"S'i a fausars et quarriaus empené

Et pic et mache et coutiel afilé,

Et dars molus avoit a grant plenté. (1305)

III. grans plomees a deriere endossé,

Puis prist se loke, e le vous adoubé.

I. longement ot el chief saielé,

Ja tant n'aroit tout le cors desmembré."

"Li Sarr. Avoit a sen costé

Les .III. espees don't I je vous ai conté.

Se grant plomee'a d'encoste torsé

Et sen picois de brun achier tempré

Miséricorde a çaint a sen costé; (2060)

En sen dos furent si .m. hauberc saffre."

"Quant .R. vit celui escapé

Hors de le tiere, molt en est aïré.

A se plomee avoit se main jeté (2270)

Que bien pesoit .i. grant caisne ramé:

IV diauble sont o lui empené."

One of the more interesting medieval manuscripts depicting war flails is the *Psalter Kupferstichkabinett* 78 A 5 (Figure 21), believed to originate from northern Italy c. 1175, though now located in Berlin (Peterson 1987; Augustyn 1989). This piece depicts war flails in no fewer than three separate folios (fols. 10r, 45v, and 52r), one in the hands of a maille-armoured and shielded figure, and two in the hands of unarmoured figures. All three flails are depicted with spiked heads attached to the haft via a chain, though the proportions between the haft and chain are slightly variable across the depictions. Two of the depictions illustrate the haft with a spiral pattern running up it (in a similar manner to some warclub depictions in the Maciejowski bible), while the third is plainer in design. These folios depict various biblical scenes with a common thread of the old testament King David, who himself is not noted to be wielding the flail in any of the scenes.

War flails are not restricted to textual, drawn, or carved depictions; several archaeological finds of flail heads have been discovered, such as the copper alloy flail head found in Flintshire, Wales, UK (Veninger 2015; https://finds.org.uk/database/artefacts/record/id/630343, accessed on 25 October 2023, Figure 22). This has been dated between 900 and 1400, though the style matches earlier Slavic flail heads widely known in Eastern Europe to originate from the 12th century. A similar flail head was recovered from a dig site in Mainz in the 1990s (Figure 23), believed to correspond to an early 13th century pattern (Taavistainen 2004; Wamers 1994).

The majority of flail heads are of the order 50–80 g, if of organic origin (bone/antler) (Michalak and Zamelska-Monczak 2016), and of a few hundred grams in weight if metal (iron, lead, brass)—much heavier, and such a weapon would be too slow and unwieldy to be a viable battlefield weapon (Zdaniewicz and Adamiak 2011). One of Sturtevant's arguments against flails being used at all was the prevalence of reproduction pieces passed off as genuine period artefacts (Sturtevant 2016a, 2016b, 2017) while completely disregarding the widespread number of finds from Eastern Europe and similar artefacts which have surfaced in the Central, Western, and Northern reaches of the continent (Taavistainen 2004; Terävä 2014). The level of decay observed on physical finds is dependent on the material of production and the inclement conditions in which they were found (Drob and Vasilache 2022). A selection of possibly misidentified archaeological finds is presented in the Appendix A.

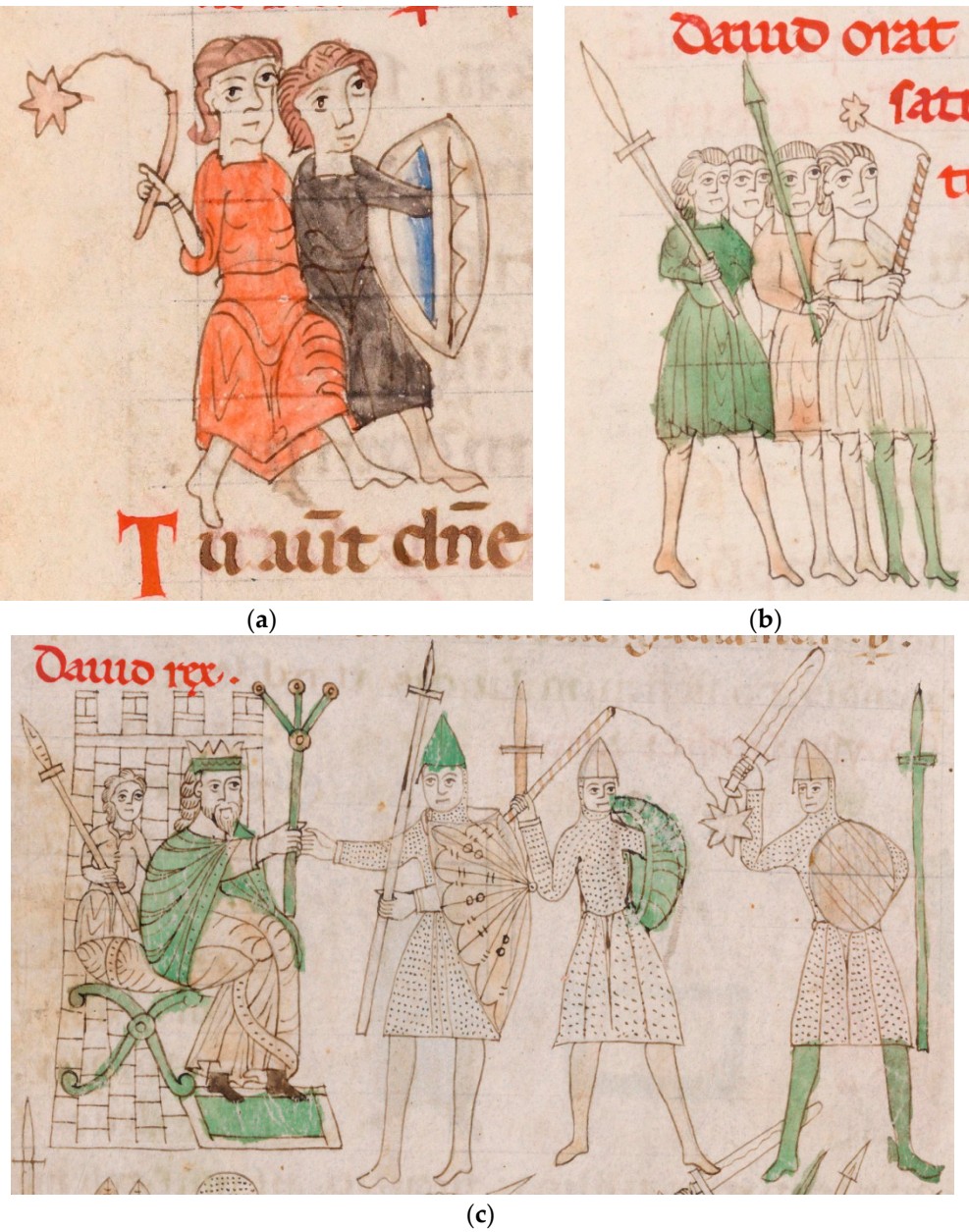

**Figure 21.** Fols. 10r (**a**), 45v (**b**), and 52r (**c**) from Kupferstichkabinett MS. 78 A 5. (Images kindly provided by Kupferstichkabinett, Staatliche Museen zu Berlin (SMB)/Kai Perstling and reproduced under the PDM 1.0 DEED license, full manuscript available digitized https://recherche.smb.museum/?language=de&question=%2278+A+5%22&limit=15&sort=relevance&controls=none, accessed on 25 October 2023).

A number of archaeological finds of "mace heads" may actually be flail heads due to the small-diameter holes being insufficient for a rigid wooden haft of appropriate thickness to maintain the necessary strength of the weapon when administering blows to the foe (Michalak 2019; Farcas 2016; Imiolczyk and Zdaniewicz 2022; Kotowicz 2008; Zdaniewicz and Adamiak 2011; Tihle 2017; Michalak 2006), despite similarities in the patterns of mace and flail heads observed (Skhorokhod and Blazhchek 2020; Daubney 2007; Michalak 2006; Florek 2019). One particular example is Figure 4.2 in the 2019 work of Michalak (Michalak 2019); in this case, the head was likely attached to a wooden handle via a piece of rope or suitably thick, or possibly plaited/braided, leather thonging knotted above and below the head (Zhirohov and Nicolle 2019; Michalak and Wolanin 2008), either of which would usually have rotted away with centuries in the ground and as such are not commonly

observed when such items are dug up (Moreno 2015); a rigid iron shaft was also present (as a mace); the remains of this would have been recovered alongside the head (Michalak 2019), and indeed wooden hafts are observed in several cases where ground conditions were favourable to preservation (Tihle 2017). A wooden haft or handle may or may not have been present on a flail (Kotowicz 2006), with the chord simply wrapped around the hand for use (Shpakovsky and Nicolle 2013; Kotowicz 2006). More examples are presented in the work of D'Amato (D'Amato 2011), though he considers these to be maces rather than flails, despite comments to the contrary in earlier writings (D'Amato 2005).

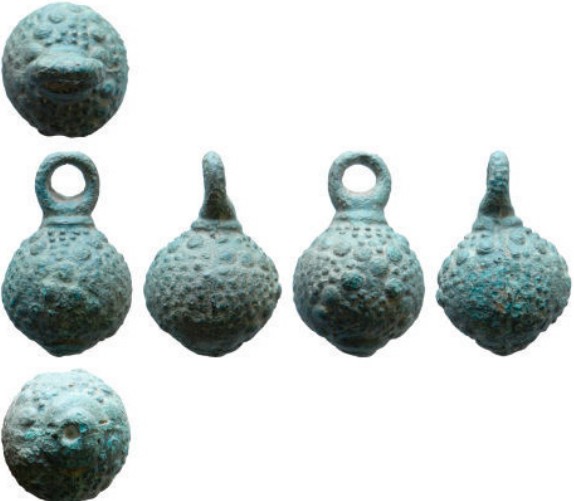

**Figure 22.** Copper-alloy flail head (Item LVPL-8F1BE0) found in Flintshire, Wales, dated 900–1400, weighs 220 g, and matches patters from Eastern Europe. (Item discussed in Veninger 2015; image courtesy of the Portable Antiques Scheme, shared under CC BY 3.0 DEED license, available with more information at https://finds.org.uk/database/artefacts/record/id/630343, accessed on 25 October 2023).

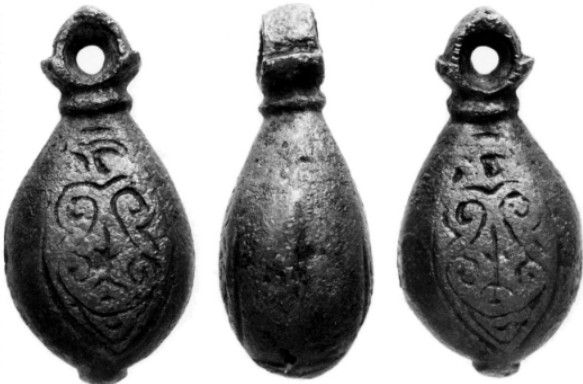

**Figure 23.** Bronze (copper alloy) weight (c. 4 cm long), identified as a flail head and matching the patterns of those found in E. Europe, found in Mainz, Germany, currently held in a private collection. (Image © Egon Wamers, permission kindly granted for non-commercial reproduction). (Wamers 1994; Taavistainen 2004) Several other examples have been found in the former territories of the Holy Roman Empire (c. 1200), which are now within the borders of Poland. (Kotowicz 2006).

A similar degree of confusion may arise between those unfamiliar with the common patterns of flail heads may arise for a range of artefacts of similar design labelled "weights". These can include the medieval "standard" weights or "steelyard weights", which bear a striking resemblance to flail head, e.g., Figure 24.

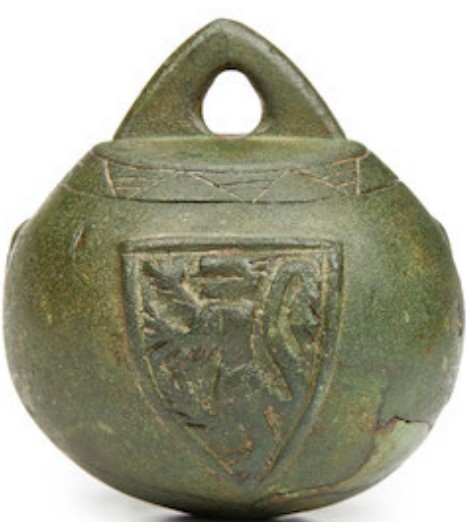

**Figure 24.** Late 13th century English steelyard weight, weighing ~ 800 g, 6 × 7 cm in size. Flail heads are typically half this size, although this form matches many several typographical matches collected from Eastern Europe. (Item was sold by Bonhams Auctioneers, London, image kindly provided from https://www.bonhams.com/auction/22681/lot/357/a-rare-late-13th-century-latten-or-copper-alloy-steelyard-weight-or-weight-of-auncel-english-circa-1272-1300/, accessed on 25 October 2023, with permission for non-commercial reproduction).

The most likely distinguishing factor between these two types of item is the overall size and weight—flail heads tend to weigh ~300 g at a maximum, with definite steelyard weights massing more than 600 g, and being correspondingly larger, though a large number identified as such are significantly lighter and could thus function effectively as flail heads. The perishability of organic materials used as the flexible component of flails mean that it would be very easy for a flail to be misidentified as a steelyard weight in this manner. An example of a lead-filled, brass shell steelyard weight is presented below. Several more possible flail heads identified as weights similar to this, though in the typical weight range of flail heads, are presented in the Appendix A.

An 1174 English seal from the reign of Henry II depicting the legendary Gnostic figure Abrasax (Abraxas) wielding a flail and shield was used to countersign a charter of Louis VII of France. This is the only known utilisation of this particular seal (Vincent 2015).

The *Chanson d'Antioche* (Song of Antioch, c. 1180) provides a similar description of the Tafurs (chant VIII, v.87, with modern French added for context, and English translation of the relevant parts), describing the use of "*plomées*" (leaden/leaded flails), and scythes—improvised weapons of the impoverished from an excerpt in the work of Funch-Brentano (Funck-Brentano 1922). A complete translated text is unfortunately unavailable to me.

"Il ne portent o [avec] els ne lance ne espée,

Mais gisarme esmolne et machu-e plomée,

Li rois porte une faus qui moult bien est temprée

N'a paien si armé en tote la contrée

K'il nel porfende tot desci qu'en la corée

Moult tient bien de sa gent la compaigne serrée,

S'ont lor sas à lor cols à cordele torsée,

Si ont les contés nus et les pances pelées,

Les mustiax ont rostis et les plantes crevées

Par quel terre qu'il voisent moult gastent la contrée."

"They do not wield lance or sword

But gisarme (farming tool)... and flail (plomée)

The King (of the Tafurs) carries a scythe which is well-soaked..."

In the last quarter of the 12th century, depictions of war flails appeared in Aragon, Northeastern Spain, for example in the archway of one of the many churches in the town of San Miguel de Uncastillo (Perratore 2015). The images depicted below (Nicolle 1999; Manning 2016) show a two-handed ball-and-chain type flail with a spherical ball and short chain (1/3 the length of the haft), though no further context of use is given. The original arch is present at the Museum of Fine Arts, Boston, USA, from which a selection of the below images (Figure 25 are taken, although a faithful copy of the archway is has been reproduced (as per Figure 26, https://www.litosonline.com/en/article/reproduction-roman-doorway, accessed on 25 October 2023).The Old French poem *Lancelot, le Chevalier de la Charrette*, written by Chretien de Troyes c. 1180, is an Arthurian tale with Lancelot as a prominent character. In this story, a giant character arms himself with a *mace de fer plomée* (flail–mace of iron) and goes on a rampage, as highlighted by the extract from the work of Huet (Huet 1912). Some adaptions of this text (Comfort 1914) fail to translate this correctly upon conversion to English.

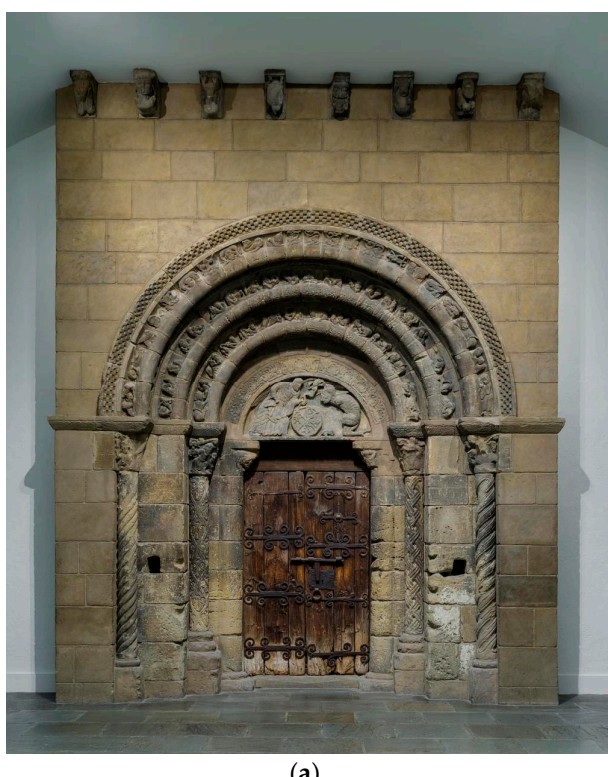
(**a**)

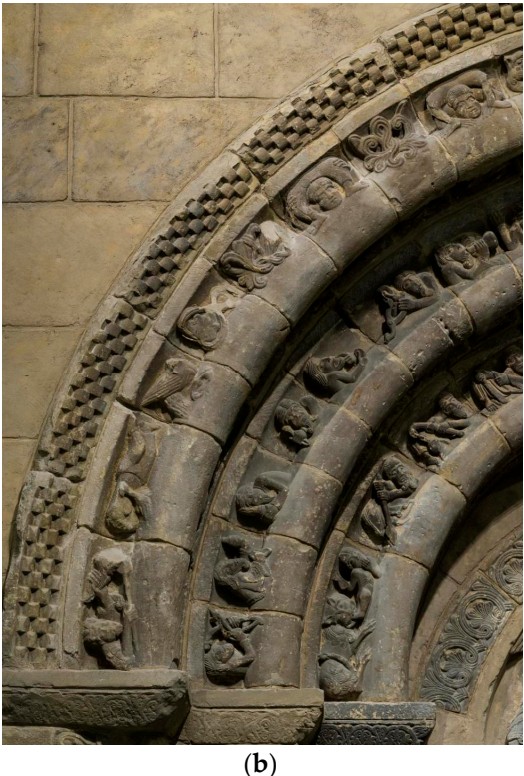
(**b**)

**Figure 25.** Carving from the entrance of the Church of San Miguel de Uncastillo, Spain; the flail is depicted on the inner ring (10 o'clock from the centre, left image (**a**); in the centre right of the right image (**b**). Item from the Bartlett Collection, number 28.32. (Photographs © Museum of the Fine Arts (MFA), Boston, MA, USA, permission granted for non-commercial reproduction, images from https://collections.mfa.org/objects/61942, accessed on 25 October 2023).

The Celtic Irish tale *Togail Bruidne Dá Derga* (The Destruction of Da Derga's Hostel), belonging to the Ulster Cycle of that country's mythology, dates from the 12th century in the earliest recorded manuscripts, but the oral tradition is likely much older (Sayers 1983). In the tale, three Manx giants (Vikings or Norse?) are described as being armed with iron flails (*susta iarnae*) comprising clubs with ball-and-chain (Sayers 1983; Brown 1919). The following excerpts are taken from the translated full text available here (https://sourcebooks.fordham.edu/source/1100derga.asp, accessed on 25 October 2023):

"A blow, they give with three iron flails having seven chains triple-twisted, three-edged, with seven iron knobs at the end of every chain: each of them as heavy as an ingot of ten smeltings. Three big brown men. Dark equine backmanes on them, which reach their heels. . .

. . . Three hundred will fall by them in their first encounter, and they will surpass in prowess every three in the Hostel; and if they come forth upon you, the fragments of you will be fit to go through the sieve of a corn-kiln, from the way in which they will destroy you with the flails of iron. Woe to him that shall wreak the Destruction, though it were only on account of those three! For to combat against 'them is not a 'paean round a sluggard."

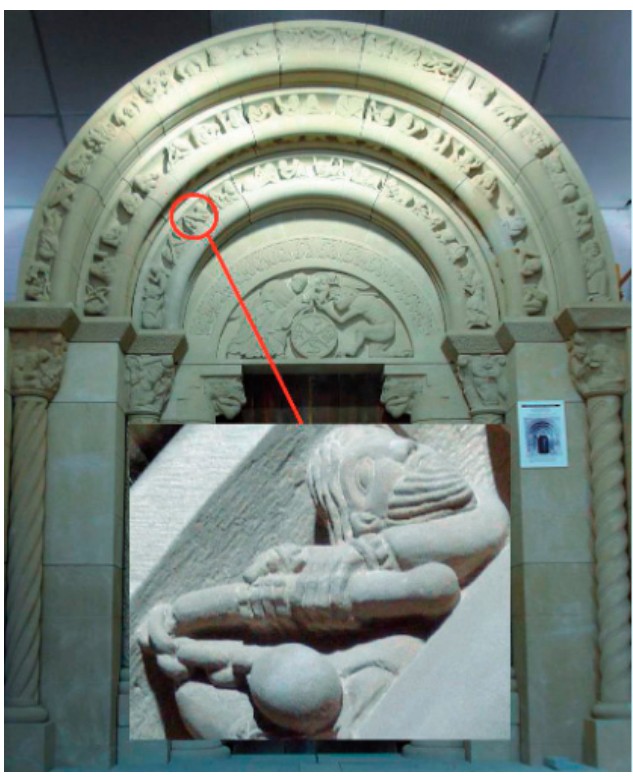

**Figure 26.** Highlighted flail on the reproduction of the original arch. (Image reproduced with kind courtesy and permission of HRM Ediciones (https://www.hrmediciones.com/), Historia Rei Militaris, vol. 8 (Moreno 2015)).

The late 12th century French chanson *La Mort Aymeri de Narbonne* (full text available at https://fr.wikisource.org/wiki/Anonyme_-_La_Mort_Aymeri_de_Narbonne/Texte_entier, accessed on 25 October 2023 and https://gallica.bnf.fr/ark:/12148/bpt6k97721205, accessed on 25 October 2023 Lines 2680–2690) features a reference to a *plomée* being used by a Christian to attack a Saracen (Moorish) camp, likely corresponding to the time of Charlemagne as many of the contemporary pieces are (du Parc 1884):

"D'une plomée va crestiens tuant,

Ça .ii., ça .iii. les ala craventant.

Et il s'escrient : « Aïde, Guiberz frans. (2685)

« Sainte Marie ! ja serons recreant. »

Guiberz l'oï, cele part vint corant,

Et li paiens s'adrece vers l'enfant ;

De sa plomée va la verje hauçant

Que la mace ert par terre traïnant,

Va la corroie larjement estendant"

"With a plomée (flail) goes Crestiens killing

First two thenIee, craving. . .

. . .from his plomée goes the heightening [strike?]

As the mace (head) drags along the ground."

The c. 1185 epic poem *Aliscans*, originally written in the Picard language of northern France (full text available at http://www.rialfri.eu/rialfriPHP/public/testo/testo/codice/rialfri%7Caliscans_%7C001, accessed on 25 October 2023 and https://archive.org/stream/aliscans00willgoog/aliscans00willgoog_djvu.txt, accessed on 25 October 2023), contains several references to war flails, referred to as both *morgensterns* and *kriegsflegels* in a modern German translation of the text (Knapp 2013). Borel, a character from the poem, is described as causing great slaughter with a war flail even against armoured foes, while another character, Margot, is described as using a morning star (Buridant 2003; Hamilton 1911), and with it killing two men with a single blow. The title of the poem is possibly a reference to the Roman settlement of Alsycamps, which continued in use until the early middle ages, and the story is likely a fictionalised retelling the story leading up to the expulsion of the "Saracens" (likely Moorish, as the term Saracen was used to refer to Arabs generally (Daniel 1979)) from southern France around the end of the 10th century. The "Saracens" in particular are noted as using a range of unusual arms including flails, a hammer, and a scythe (Schultz 1889), though these kinds of descriptors are not unique to *Aliscans*: many of the other *chansons* portray the "Saracens" with different weapons, likely as a means to emphasise their "otherness" (Lofmark 1972; Hamilton 1911). The story is believed to be based upon the earlier *Chançun de Willame*, and inspired the later *Willehalm*, composed by the German knight Wolfram von Eschenbach (Gibbs and Johnson 1997). The latter of these also cites the use of war flails (*kriegsflegels*). The effectiveness of flails against maille-armoured opponents is discussed later, but this is recognised in *Aliscans*, thus likely reflecting period knowledge of the use of these weapons in a combat or battlefield setting. Flails are described around lines 5719, 5739 (where the Count Guillaume d'Orange flees a flail-wielding opponent, possibly a pagan), and 5990 (Sternberg 1886):

"Un flaiel porte, la mace ert d'orpuement

Et tout li mances en estoit ensement

Et la chaine don't la batier pent

Plain poig ert grosse, close estoit fierement

Ki ert molt dure, d'une pel de serpent

Ki ne crient arme d'acier ne ferrement."

"N'i a celui ne portast I flael

Toz sont de coivre, bien over a cisel."

A flail is described in the c. 1200 German poem *Nibelungenlied* or *The Song of the Nibelungs* which is believed to originate from Passau in the south of the modern-day Germany (Jähns 1899; Whobrey 2018; Shpakovsky and Nicolle 2013). The duel between the character Alberich (Albrich) and a "dwarf king" is described as follows (from a modern-day translation (full text available at https://www.gutenberg.org/files/7321/7321-h/7321-h.htm, accessed on 25 October 2023)):

"Alberich was full wrathy,/thereto a man of power. (v. 494)

Coat of mail and helmet/he on his body wore,

And in his hand a heavy/scourge of gold he swung.

Where was fighting Siegfried,/thither in mickle haste he sprung.

Seven knobs thick and heavy/on the club's end were seen, (v. 495)

Wherewith the shield that guarded/the knight that was so keen

He battered with such vigor/that pieces from it brake.

Lest he his life should forfeit/the noble stranger gan to quake."

The weapon described here is referred to as a *giessel* (whip) in the original text, though this seemingly describes a war flail, though a bit of artistic license on the part of the authors is likely here, unless the reference to gold is instead describing copper alloy in the form of brass or bronze which can appear golden. Most of the period depictions of war flails picture these with single heads, rather than the seven attested to in the poem, with multi-headed flails only appearing later (post 1250), barring a few exceptions previously highlighted. Shpakovsky and Nicolle postulate that the Germans viewed this as an alien weapon of some rarity in their locale, but nonetheless an awareness of the technology is demonstrated (Shpakovsky and Nicolle 2013). Late 13th century surviving versions of this work are extant and can be found https://www.digitale-sammlungen.de/en/view/bsb00035316?page=1, accessed on 25 October 2023.

The c. 1200 French chanson *Jourdain de Blaivies* (full text available at https://archive.org/details/amisetamilesundj00hofm/page/218/mode/2up?q=quars, accessed on 25 October 2023), which tells a tale of revenge in the time of Charlemagne, describes the use of a war flail (*plomée*) (Schultz 1889; Sternberg 1886):

"Li uns pleut hache et li autres espec (3968)

Li tiers sa mace et li quars sa plommée."

A 1202 inventory of weapons prepared after the capture of Robbio Castle (northern Italy) included, amongst other things, *falciones* (falchions/fauchards) and *plumbatas* (*Kriegsflegels*— war flails), though no further description than this is provided, and the original source (Angelucci Documenti) is unavailable (Köhler 1887).

The French epic poem *Huon of Boreaux* (c. the first half of the 13th century) describes a confrontation between the titular character and a pair of "men of brass, each of whom held in his hand an iron flail… beat with their flails without ceasing for a single moment". These "brass men" are guarding a tower in which a damsel named Sybil is imprisoned by the giant Angolafer (perhaps the origin of so many "princess in the tower" stories). Huon triumphs over the brass men and slays the giant (Church 1904; Tuve 1929). The legend itself is believed to originate from the time of Charlemagne. The full text is unfortunately unavailable to me.

The *Vercelli Mappamundi* (Vercelli Map of the World, c. 1217, held in the Archivio Capitolare of Vercelli) (Mittman 2019), believed to be either an early or later 13th century stylised map of the world, depicts a figure carrying a multi-headed flail (more likely than a scourge, Figure 27). The accompanying text describes this crowned figure, who is riding a stylised giant bird bearing a horseshoe (described in one reference as "the iron-eating ostrich") in its beak as "King Philip of France", which could refer to Philip II (1165–1223) or Philip III (1245–1285) (Vercelli Mappamundi n.d.; Davies 2020; Lewy 2021). Although otherwise heavily damaged, this portion of the map is still legible.

Thorbeck's interpretation of the events of the Battle of Las Navas de Tolosa (1212), where the power of the Moors during the Reconquista was broken, sealing their ultimate demise several centuries later, describes the use of a "*látigo de guerra*" (lit. war whip cf. Schlachtgeissel) by the Navarrese king Sanco the Strong, perhaps based on the account of the troubadour Guillermo de Aneliers (Thorbeck 1999; Moreno 2015; Arnedo 2010). The flail used or a similar example thereof was supposedly interred within the Abbey of Roncesvalles (Thorbeck 1999; Moreno 2015; Arnedo 2010), though items found are also attributed to the flails of Roland and Olivier (Demmin 1911), but these may be later reproductions or forgeries designed to act as "holy relics", as these were first mentioned in a note from 1617 (Dona 2021). The original Thorbeck reference is unavailable, however, and as such the sources he used cannot be analysed in any further detail. The remains of a flail were purportedly found upon the field of Battle of Las Navas de Tolosa (Abad 2013; Moreno 2015), though again, this text is not available and so cannot be discussed or analysed further.

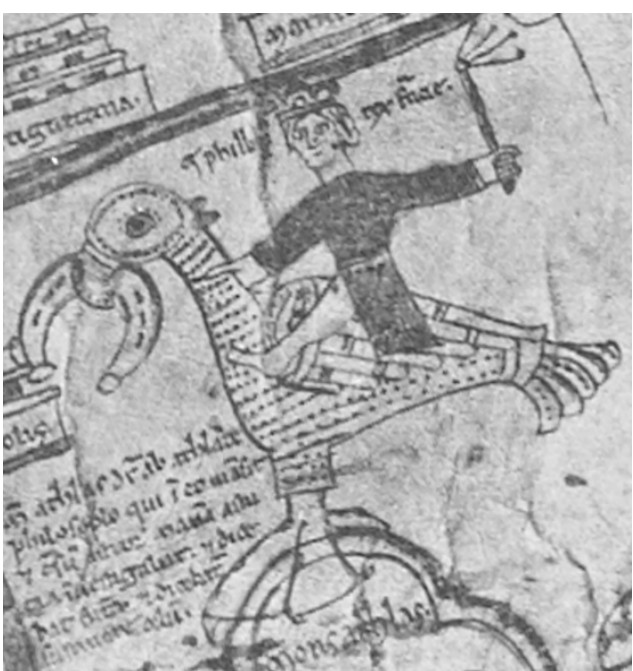

**Figure 27.** Depiction of King Philip II or III of France from the *Vercelli Mappamundi* (© Fondazione Museo del Tesoro del Duomo e Archivio Capitolare di Vercelli, Italy, permission granted for non-commercial reproduction, Mappamundi, Capitulary Archive—Vercelli, Rotoli figurati, 6).

(XXXIV) A series of paintings from Iglesia de la Asunción, Alaiza, Spain depict several figures (perhaps demons?) armed with ball-and-chain type flails (Figure 28). The church itself is dated from the early 13th century, though the style of the artwork (dress worn by the figures therein) more reflects late 12th century styles.

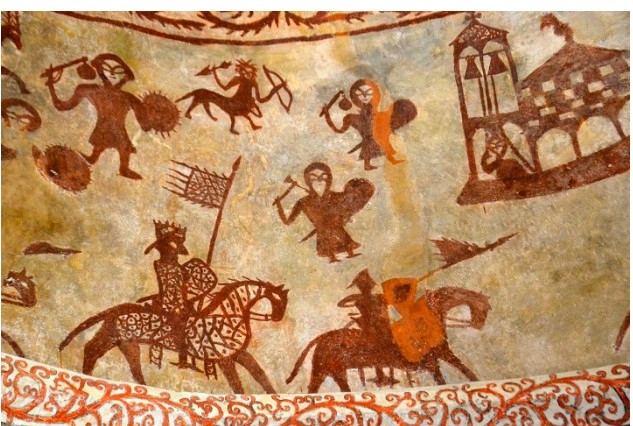

**Figure 28.** Paintings from the church of Iglesia de la Asunción, Alaiza, Spain. (© Asier Sarasua Aranberri, image sourced from https://www.flickr.com/photos/asarasua/28624745413/, accessed on 25 October 2023 under CC BY-SA 2.0 DEED license).

The Roman treatise *De Re Militari*, originally written in Latin, was translated into French and Norman French during the early and mid-13th century, respectively (as the *Fitzwilliam Vegetius*) (Carley 1962; Moreno 2015), and includes many references to the use of flails in a military setting. The original Latin text was adapted during the translation process to account for period terms (Carley 1962; Moreno 2015). Due to the number of references to the flail (as *plomez* or *plommez*), I will refer readers to the thesis of Carley (Carley 1962) for further discussion as the relevant portions of the text are discussed in detail.

The 13th century French chanson *Doon de Mayence* mentions the use of the *plomée* (flail), M 10,624 (Sternberg 1886). The full text of is available at https://archive.org/stream/bub_gb_wXY-8Njcsg4C/bub_gb_wXY-8Njcsg4C_djvu.txt, accessed on 25 October 2023.

"Dont li veissies pierres et pessemens ruer Et de lanches ferir et d espees capler

De maches de plommees merveilleus cous donner"

The stained glass windows of Chatres Cathedral (France, constructed around 1160), most of which are believed to date from the first half the 13th century; bay 100 (c. 1230–1250) depicts the 2nd century martyrs Saints Gervais (Gervasius) and Proteus (Protasius) (Kolar 2020), who is carrying a book in his left hand and an inverted ball-and-chain flail in his right, the haft held to his chest (Figure 29). This bay appears adjacent to the bay (99) depicting Saints Damien and Cosmos (https://www.therosewindow.com/pilot/Chartres/w118.htm, accessed on 25 October 2023). The flail represented is depicted with a long red haft, perhaps intended to represent wood, with silvery fittings (iron) connecting a brown (copper alloy) head via a short chain. This represents one of the clearest representations of the fittings of a flail to its haft which, period artistic license notwithstanding, would suggest a conical socket containing a ring of iron, to which the chain of the flail is attached. This type of construction was well within the capabilities of smiths in the 13th century and indeed earlier. The flail may also be an unusual depiction of a threshing flail (given the shortness of the item), as the saints depicted are the patrons of haymakers (Kolar 2020).

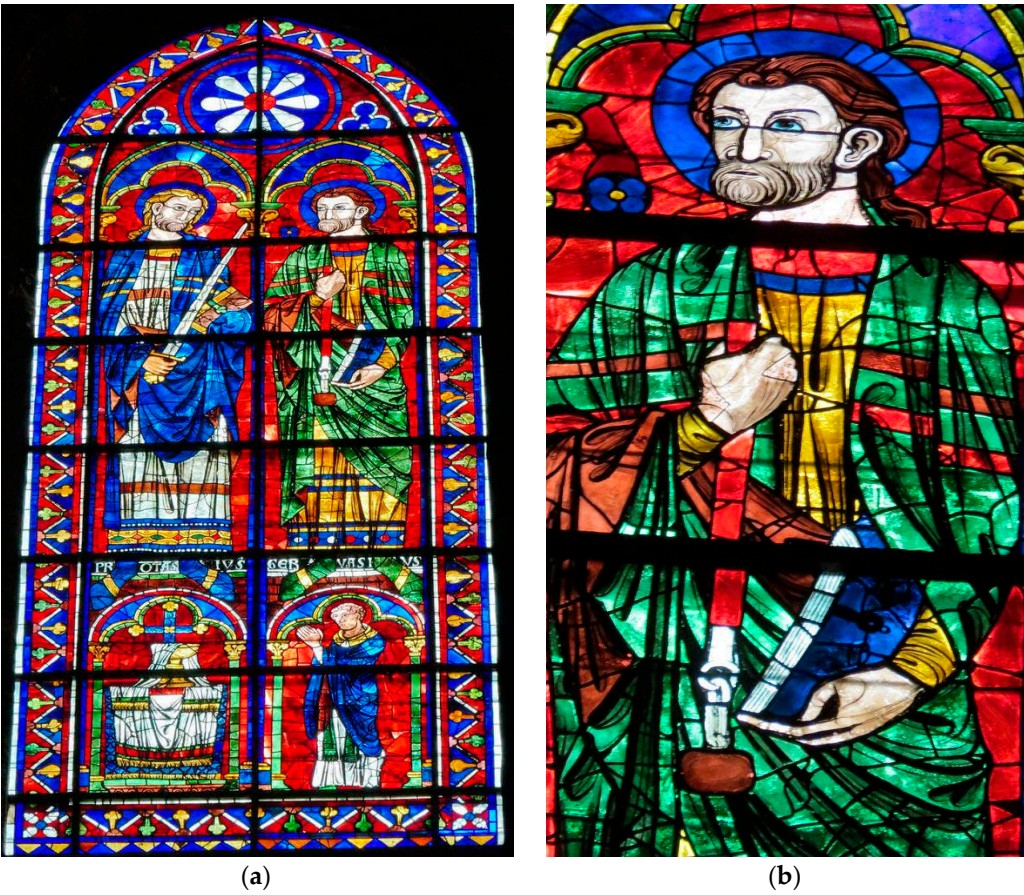

(**a**)  (**b**)

**Figure 29.** Stained glass bay 100 of Chatres Cathedral (**a**) with a zoom on Proteus (**b**). (© Jean-Yves Cordier, kindly provided with permission for reproduction, images from https://www.therosewindow.com/pilot/Chartres/w118.htm, accessed on 25 October 2023).

The c. mid-13th century French or English manuscript *Bible Moralisée* (dated 1226–1275), currently held in the Bodleian Library at the University of Oxford (MS. Bodl. 270 b, Figure 30) contains a great many folios depicting demons tormenting people with an

assortment of odd weapons, including a ball-and-chain flail (fol. 156r). In this image, a pink-hued demon (distinguished by horns, pointed ears, and a tail) wields a war flail two-handed, reinforcing the common iconography of flails being weapons wielded by the evil. The haft is depicted in brown (wooden) and the chain and (possibly spiked) ball in grey, suggesting construction in iron. The construction of this flail is one of the clearer examples presented and mirrors the design of the one presented above in several respects. This represents one of the later depictions of flails being used in this setting.

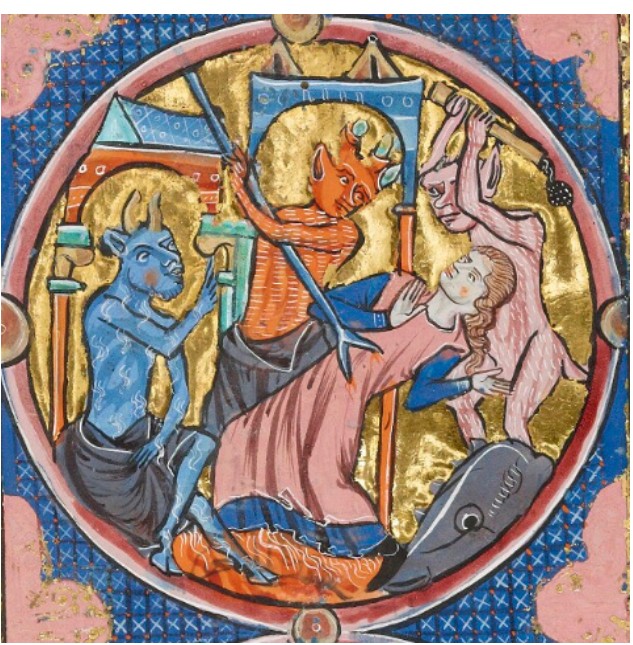

**Figure 30.** Fol. 156r from the Bible Moralisée (© The Bodleian Libraries, University of Oxford, MS. Bodl. 270b, permission granted for non-commercial reproduction under CC BY-NC 4.0 license, image from https://digital.bodleian.ox.ac.uk/objects/4cf7e9d2-c06e-4029-a3b1-152736320897/surfaces/9ab402ec-72dd-4cbb-ac98-d9e43f7a0330/, accessed on 25 October 2023).

*Later Sources*

The number of visual depictions of war flails increases over the course of the 13th century and onwards, with evolutions in style and use as will be discussed in due course, as the adaption of threshing flails into a weaponised form during the 14th century was presented above. From c. 1250, war flails are depicted more commonly in use by heavily armoured, mounted warriors rather than unarmoured or more lightly armoured dismounted figures in earlier artwork. As such, a selection of pertinent examples from after 1250 is presented, rather than all known sources. A similar transition further takes place from this point in time, when the previously numerous depictions of flails being wielded by "evil" persons (heathens, mobs, heretics) and creatures (demons) is surpassed by knightly or chivalric examples.

One exception to this observation is fol. 179 from the Radziwill Chronicle (MS 34.5.30, currently held in the Research Department of Manuscripts, Library of the Russian Academy of Sciences, St. Petersburg, Russian Federation, Figure 31), depicting the murder of Prince Ihor (Igor) by the Kyivans during the 1147 uprising (Taavistainen 2004).

The manuscript itself is dated from the 15th century, but believed to be a copy of an earlier 13th century manuscript. A short war flail is depicted being wielded by the figure in red—a wooden haft with a blue head (Osypenko 2020; Sturtevant 2017). This is likely more representative of the flails used in Eastern Europe rather than those found more westwards where multi-headed designs become more prevalent after this point in time.

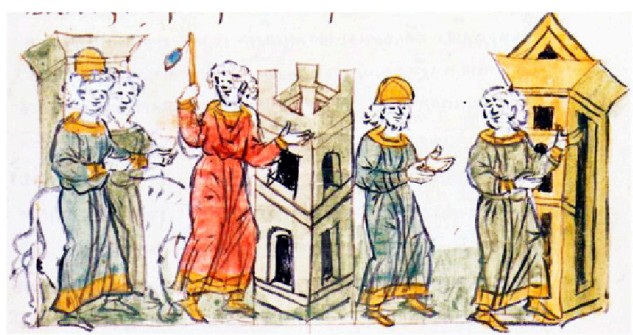

**Figure 31.** Fol. 179 from the Radziwill Chronicle. (© Library of the Russian Academy of Sciences, MS 34.5.30, permission granted for non-commercial reproduction).

One of the earliest depictions of war flails being used by mounted warriors in Western Europe is a folio (210v) from the History of Outremer (BNF Français 2630, France, Figure 32), dated c. 1250. A single- or possibly multi-headed iron flail with a wooden haft is depicted being used by a mounted warrior in the top left of the image. Given the usual "lag" in the visual reporting of items being depicted in comparison to their actual use, this serves as confirmation that war flails were indeed in at least sporadic use in Western Europe at least as early as 1250, and very likely before this.

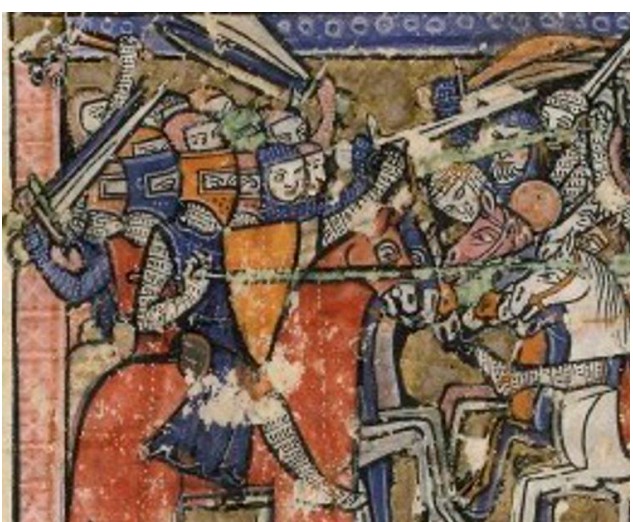

**Figure 32.** Fol. 210v from the History of Outremer (© Bibliothèque nationale de France, BnF MS 2630, permission granted for non-commercial reproduction, image from https://digital.bodleian.ox.ac.uk/objects/4cf7e9d2-c06e-4029-a3b1-152736320897/surfaces/9ab402ec-72dd-4cbb-ac98-d9e43f7a0330/, accessed on 25 October 2023).

The Count of Holland and Zeeland, William II (1227–1256), was reportedly killed by a blow to the head from a flail (type unspecified—referred to in the Dutch text as a Strijdvlegel) after his horse fell through ice on a frozen lake during the Battle of Hoogwoud (modern-day Netherlands) against the Frisians. His body was buried under a house but recovered by his son and then reburied in the Abbey of Middelburg (Netherlands) (Burgers 1998).

An inventory of the armoury of Sesa Castle (Huesca, Aragon, Spain) from 1274 describes the inclusion of "3 iron maces with their chains", (listed as *iii macas de fierro con sus cadenzas*). As the Spanish word "maca" is the standard description for a ball-headed mace, this entry likely refers to flails rather than other possibilities.

A circular boxwood artefact (c. 4 cm in diameter, Figure 33) believed to be a draughts or backgammon piece originating from England depicts an armoured solider attacking what appears to be a giant, helmeted mollusc with a war flail. The timeframe of this piece

can be narrowed by the ailettes present on the shoulders of the figure—items of armour which only came into brief use in the latter half of the 13th century and were replaced not long after the turn of the 14th century by dedicated shoulder cops and later pauldrons as metallurgy improved over time. The flail being wielded appears to be a ball-and-chain type. This item is held in the collection of British Museum.

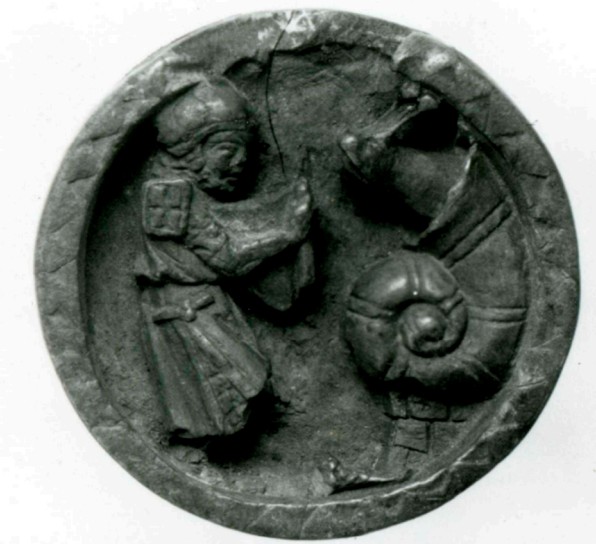

**Figure 33.** Draughts piece from the British Museum (Item 1853,0214.2). (© The Trustees of the British Museum, shared under a Creative Commons Atttribution-NonCommercial-ShareAlike 4.0 International (CC BY-NC-SA 4.0) license for non-commercial reproduction only, image from https: //www.britishmuseum.org/collection/object/H_1853-0214-2, accessed on 25 October 2023).

A folio from *Moniage de Guillaume* (France), dated c. the late 13th century, depicts two dismounted, maille-armoured warriors, one armed with a three-headed war flail, and the other with a giant two-handed war club and a sword belted at his waist. This image may have connections to the symbology of the 12th century Waldensian movement of France, whose leader is connotated with wielding a spiked flail (Gerhardt 2007), or perhaps represent Guillaume and Rainouart from the Cycle of Guillaume, discussed previously. The folio can be found at: https://www.meisterdrucke.de/kunstwerke/1200w/Italian_School_-_ Warriors_arms_of_clubs_-_in_The_moinage_of_William_manuscript_of_the_13th_centu_ -_(MeistferDrucke-970716).jpg, accessed on 25 October 2023.

A series of frescoes from a late 13th—early 14th century Italian church (now destroyed) depict the story of *La Chanson d'Otinel*, a late 12th century Norman or French tale about a Saracen (likely Moorish) knight from the time of Charlemagne who converts to Christianity. The titular character is depicted fighting on foot wielding "a flail or with three chains ending in lead spheres", in contrast to the typical text of the story, where he fights mounted with a sword (Gerhardinger 2020). As Roland and Olivier (from the *Song of Roland*) are somewhat intertwined in this story (Fein and Raybin 2019), the character depicted in these frescoes may be Olivier, as per the fresco from the Cathedral of Verona (Figure 19).

The c. 1277 French treatise *De Regimine Principum*, written by Giles of Rome, serving as Archbishop of Bourges (c. 1243–1316) for King Philip the Fair of France provides instruction on the correct governance of a kingdom in several sections including warfare, and was translated into several contemporary languages. The flail is explicitly mentioned (with reference to ball and chain) in both the French and Latin versions as *plomée* and *plumbatis*, respectively (Scala 2021). To paraphrase both approximate translations: "For a ball of lead or iron, when some chain is attached to the wooden handle it gives a strong blow. For the sake of a ball with a chain attached to the spear strikes the air more violently than if they were themselves the spear itself would be attached to a wooden handle".

Folio 282v from *Arthurian Romances* (Beinecke MS.229, France, Figure 34), dated c. last quarter of the 13th century, depicts a maille-armoured figure wielding a three-headed flail (less likely a scourge, given a single head per strand). Somewhat amusingly, he is riding (or possibly choking) a giant chicken (Shartrand 2020), suggesting that a degree of artistic license is being taken here. The artist likely intended to convey a wooden-handled flail with iron heads. Whether these are connected to the haft via chains or another method cannot be determined. This may be a reference to King Philip the Fair of France (c. 1285–1314) or Guy of Dampierre (Shartrand 2020).

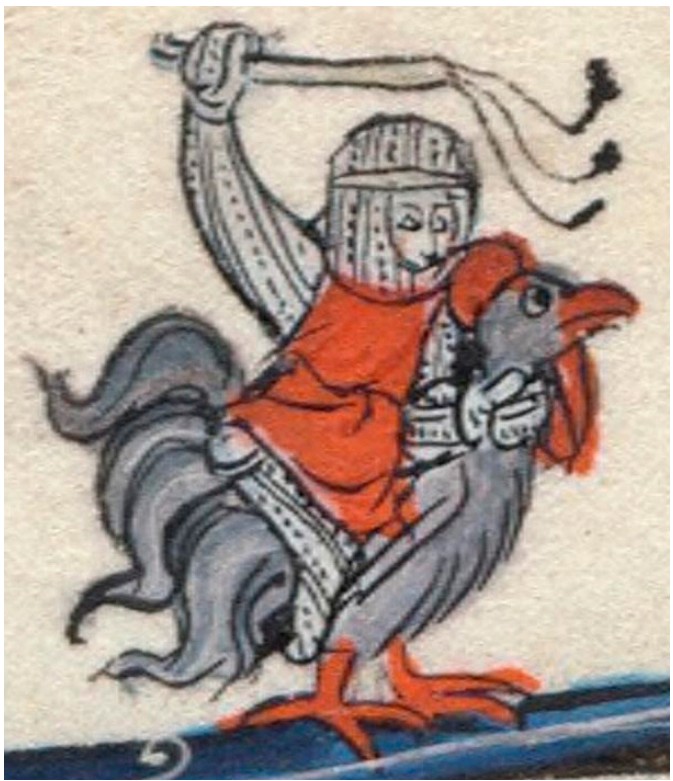

**Figure 34.** Fol. 282v from *Arthurian Romances*. (Provided courtesy of the Beinecke Rare Book and Manuscript Library, Yale University, New Haven, CT, USA, image from https://manuscriptminiatures. com/3971/10513, accessed on 25 October 2023).

The French poet Guillaume Guiart (d. 1316) wrote of the use of flails (as *plomées*) in *Branche des Royaux Lignanges*, covering the Franco-Flemish war (1297–1305), where he was injured in 1304. The references are as follows (Schultz 1889), lines 1469 and 6948:

"La véissiez enteser maces

Et plomées pour faire plaies."

"La ot tant bastons et plomées."

A folio from (131v-2) from the early 14th century French manuscript *Vita et Passio Beati Dionysii* (BNF Latin MS 5286, Figure 35) depicts a large battle scene where one of the soldiers (centre top) is wielding a ball-and-chain-type war flail. This appears as a two-handed weapon where the chain or chord is about half the length of the haft. As the image is black-and-white and the wielding character is in the background, however, further information cannot be garnered.

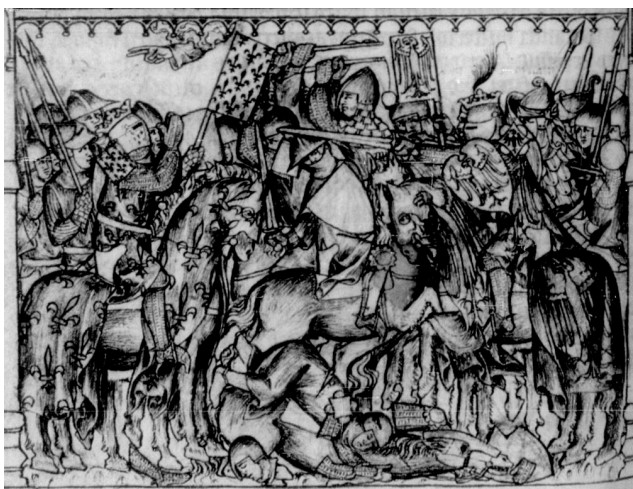

**Figure 35.** Fol. 131v-2 from *Vita et Passio Beati Dionysii*. (© Bibliothèque nationale de France, Gallica, BnF MS 5286, permission granted for non-commercial reproduction, image from https://manuscriptminiatures.com/5870/23223, accessed on 25 October 2023).

The c. 1310–1340 Franco-Italian manuscript *Libreria Marciana* Fr. Z 21—*Entrée d'Espagne* contains no fewer than thirteen folios (fols. 17v, 22v, 23r, 23v, 24r, 27r, 32v, 33v, 36r, 43v, 44r, 44v, and 47v) of a mounted (possibly Saracen) knight (described as Farragu, or Farragut) wielding a three-headed ball-and-chain flail which in some of the folios is also presented with an additional spherical "head" at the top of the haft. The folios seem to have been illustrated by at least two different artists given the variance in design between the first (17r–27r, Figure 36) and second (32v to 47v, Figure 37) sets of similar folios from the manuscript. These two groups of folios are presented below, with a description (describing the flail as a "3-stringed mace") available from https://www.rialfri.eu/en/manoscritti/venezia-biblioteca-nazionale-marciana-str-app-21, accessed on 25 October 2023.

The (Saracen) knight (paladin) depicted with the flail (Farragut) Is one of the characters within several texts of the Matter of France and some Spanish texts dating back to the mid-12th century at the latest, in addition to appearing in some Italian epics. He is sometimes depicted or referred to as a giant, and was eventually killed by the aforementioned Roland, who is also described in the manuscript from which these folios are drawn, alongside Charlemagne. Many of the other depictions of Farragut show him armed with more conventional knightly weapons (lance, sword), rather than the flail, but this is consistent in Fr. Z 21. The flail depicted is undoubtedly effective in the story as several of Farragut's opponents are unhorsed by him (fols. 22v and 27r).

An "Arming Treatise" entitled the *Manner of Arming Knights for the Tourney* (f.86v and 87 from BL Additional MS 46919), believed to have been compiled by Fr William Herebert of Hereford (d. 1333 or 1337) in the early 14th century, describes flails in the equipment of knights during tournaments (Moffat 2010). The type of flail mentioned here is not specified, but this suggests that flails were at least used in tournament settings. Note the use of the Latin-derived term "*flagellu*", rather than the modern French "*fléau*", which posed challenges in translating this piece (Moffat 2010).

> "ayne payns ou gayns de baleyne sa espeye i. gladi' & flagellŭ & galeam i. heaume."

> "gaignepains or gauntlets of baleen, his espeye that is sword, and flail, and helm that is heaume."

A small, single-headed ball-and-chain flail is depicted in a folio (049r, Figure 38) from the c. 1330–1340 illuminated French manuscript Avignon BM MS.121 *Psalter-Hours* in the hands of a maille-clad dark-skinned warrior who also bears a small round shield, perhaps a buckler. A number of similarly armoured figures wielding an assortment of polearms and a lantern are present in the background, perhaps mirroring the mob arresting Jesus in the 12th

century frescoes presented above. The artist has chosen not to represent the wooden hafts of the various weapons present with brown pigment but instead coloured these in grey, as per the maille and steel or iron of the weapon heads, despite there being brown present in sword scabbards and hair. The flail depicted is one of the shorter examples known.

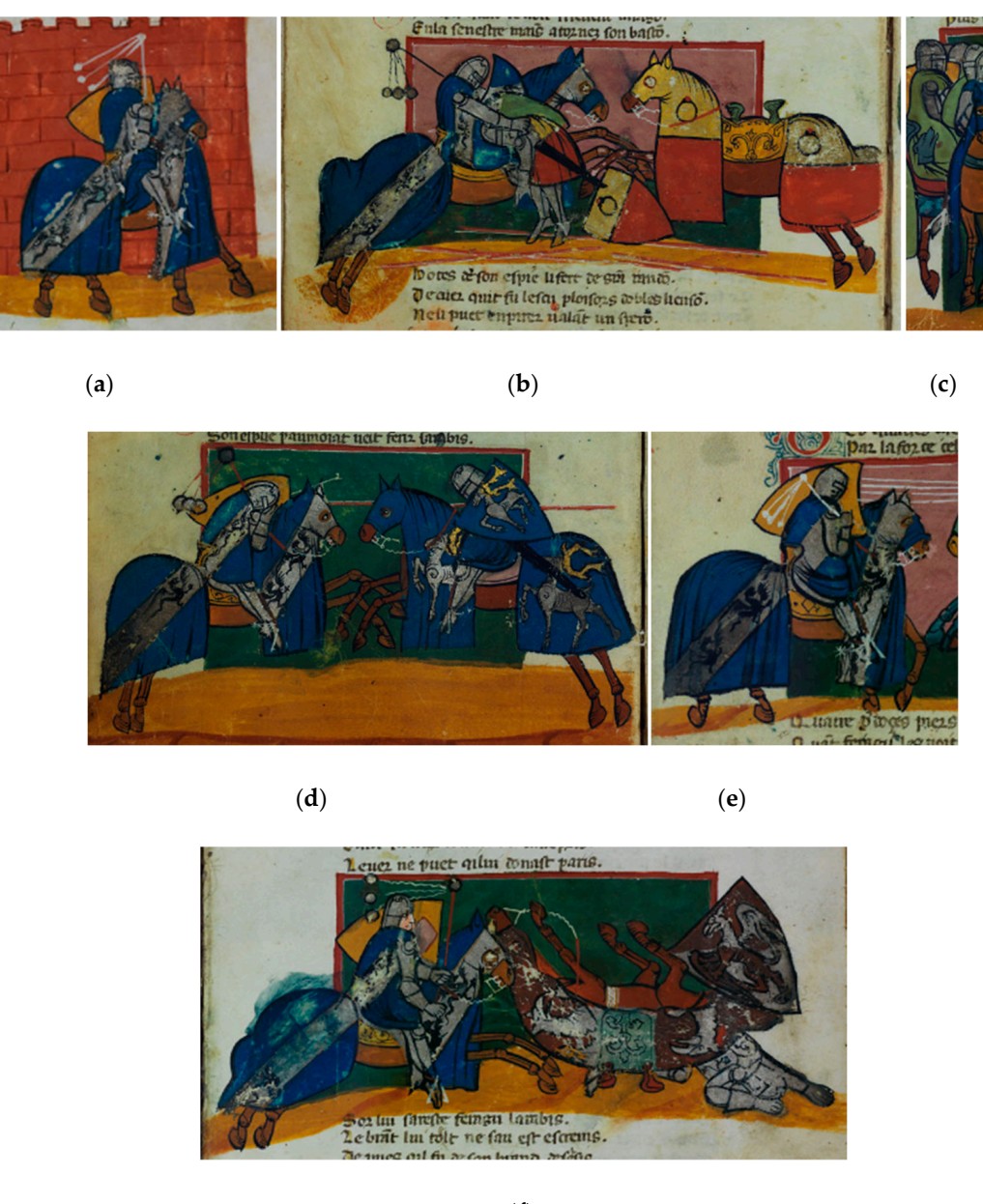

(**a**)                    (**b**)                    (**c**)

(**d**)                    (**e**)

(**f**)

**Figure 36.** Fols. 17v (**a**), 22v (**b**), 23r (**c**), 23v (**d**), 24r (**e**), and 27r (**f**) from *Libreria Marciana*. (© Biblioteca Nazionale Marciana (Venice)/ICCU (Istituto Centrale per il Catalogo Unico)/RIALFrI (Digital Repository of Medieval Franco-Italian Literature), Italy, MS francese Z 21 (257) with permission for non-commercial reproduction under CC 1.0 license. Images are from https://www.internetculturale.it/jmms/iccuviewer/iccu.jsp?id=oai:193.206.197.121:18:VE0049:CSTOR.243.15237, accessed on 25 October 2023).

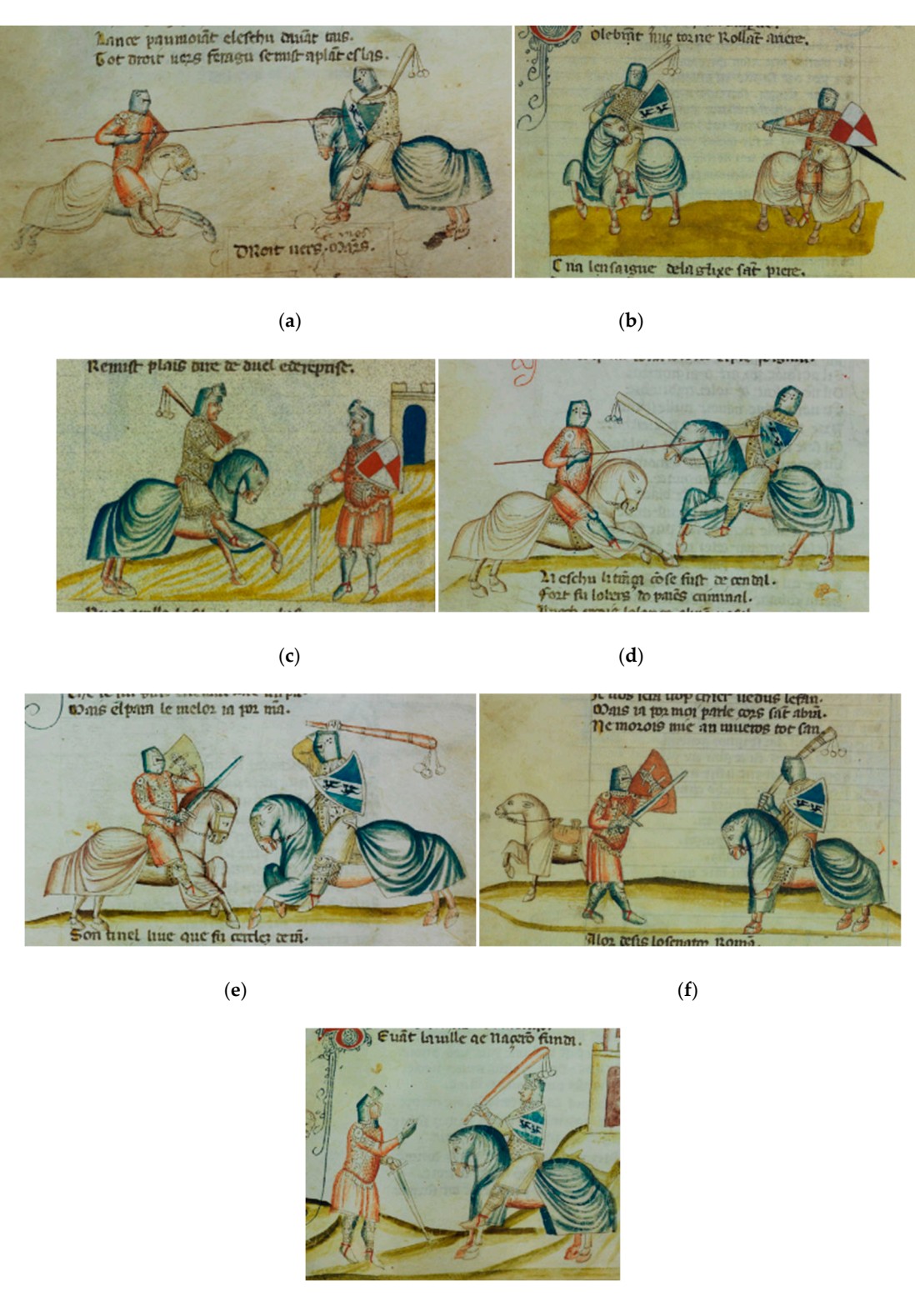

**Figure 37.** Fols. 32v (**a**), 33v (**b**), 36r (**c**), 43v (**d**), 44r (**e**), 44v (**f**), and 47r (**g**) from *Libreria Marciana*. (© Biblioteca Nazionale Marciana (Venice)/ICCU (Istituto Centrale per il Catalogo Unico)/RIALFrI (Digital Repository of Medieval Franco-Italian Literature), Italy, MS francese Z 21 (257) with permission for non-commercial reproduction under CC 1.0 license. Images are from https://www.internetculturale. it/jmms/iccuviewer/iccu.jsp?id=oai:193.206.197.121:18:VE0049:CSTOR.243.15237, accessed on 25 October 2023).

In a demonstration of flails bearing symbology outside of the Christian sphere (albeit in Europe), a 14th (or possibly late 13th) century legend held by the extant Islamic (Sufi) sect known as the Bektashi, originating from Albania, told of two giants (Tomorr and Shpirag) who fought for the love of a goddess, with one (Shpirag) attacking the other with a cudgel, or flail. Shpirag is denoted as being the "evil" brother here and as such the connotation of flails being "weapons of the evil" persists across cultures and faiths (Kristo 2020).

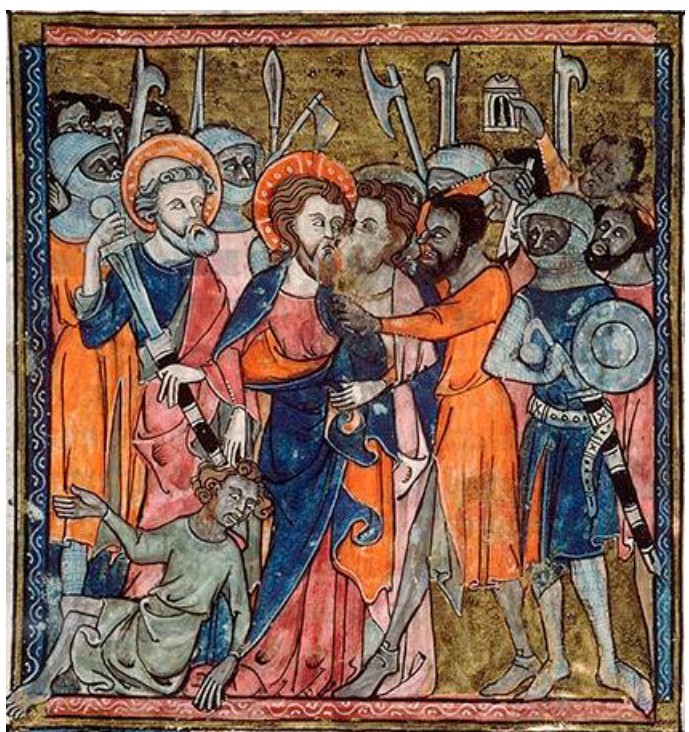

**Figure 38.** Fol. 49r from *Psalter-Hours*. (© Avignon Bibliotheque Municipale, permission granted for non-commercial reproduction, image from http://www.enluminures.culture.fr/Wave/savimage/enlumine/irht2/IRHT_055057-p.jpg, accessed on 25 October 2023).

Folio 5v from the *Life of Eustace and Other Saints* (BL Egerton 745, Paris, France, Figure 39), dated c. 1st quarter of the 14th century, depicts a battle between crusaders and Saracens, or perhaps the Moors. One of the Saracens is wielding a multi-headed ball-and-chain flail, interestingly containing three heads per strand across multiple strands. These kinds of items are normally depicted as scourges, though these are not really battlefield weapons. The colouration of the piece suggests the weapon in question has a wooden haft and iron or possibly stone balls, perhaps with an iron chain, though this is not pictured.

Further descriptions of flails appear in the same literary cycle describing the battle where Finn is killed (See (I)) (Battle of Ventry—Cath Finntrágha in Irish)—a "strong iron flail... with seven balls of refine iron, and fifty iron chains from it, and fifty apples on each chain, and fifty venomous thorns on each apple" is wielded by Mongach of the sea to attack the King of the Bretons. This was first recorded in the 15th century, but as with the rest of the cycle, the original oral tradition is much older (Gregory 1905; full texts available at https://www.maryjones.us/ctexts/f20.html, accessed on 25 October 2023 and https://www.sacred-texts.com/neu/celt/gafm/gafm45.htm, accessed on 25 October 2023).

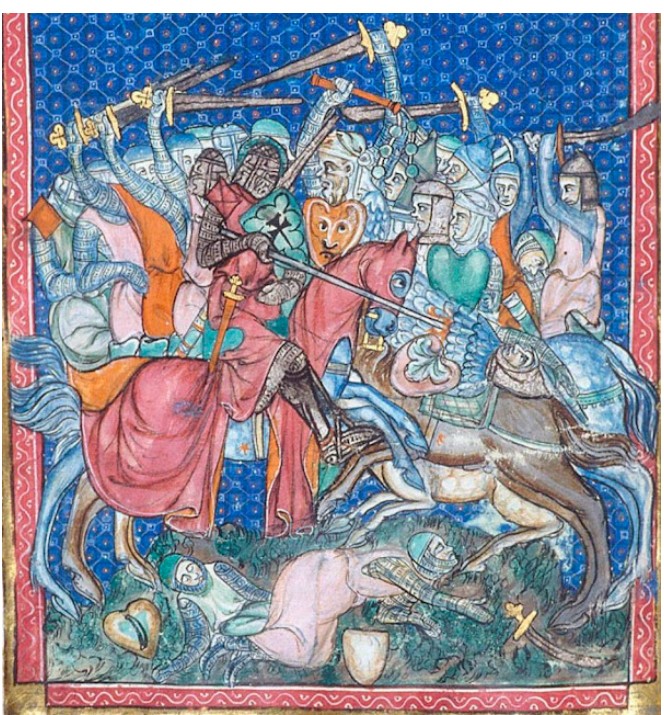

**Figure 39.** Fol. 5v from the Life of Eustace and Other Saints. (© British Library Board, BL MS Egerton 745, permission granted for non-commercial reproduction, image from https://www.bl.uk/catalogues/illuminatedmanuscripts/ILLUMIN.ASP?Size=mid&IllID=9856, accessed on 25 October 2023).

## 9. The Practical Use of Flails and their Context in Warfare in the Medieval Period

Flails fall under the category of "blunt", "impact", or "crushing" weapons, of which other weapons in the class (maces, clubs) were far more common in the medieval period (Tihle 2017; Mitchell 2002; Balbuena and Belda-Medina 2022). Such blunt weapons, clubs in particular, were often seen as weapons of the lower classes in the medieval period, and were often depicted in the hands of heathens or pagans in some artwork (Tihle 2017), although the nobles of most nations were pragmatic enough to see the benefits of such weapons in addition to their swords, spears, and lances, despite the relative lack of prestige (Moreno 2015).

In Eastern Europe, war flails were commonly used by mounted troops in the early and high medieval periods (Kirpičnikov 1968; Shpakovsky and Nicolle 2013), while as the majority of artwork presented above demonstrates, outside of this region, in Western, Central, and Southern Europe, they were predominately weapons of the infantry, at least up until 1250–1300; threshing flails and their weaponised derivatives would be purely the purview of infantry, given the impracticality of using such a weapon mounted, notwithstanding the adapted designs which found use by the Mongols (Kuleshov 2019).

In contrast to bladed weapons, which inflict cutting injuries on unarmoured flesh and bone, and lances, spears and arrows, which inflict piercing injuries, flails, maces, and clubs inflict damage through concussive or "blunt" force, which are capable of inflicting significant damage even through armour, depending upon type (Mitchell 2002). As maille (chainmail) armour, the pinnacle of armour technology up to around 1300, is flexible, the impact imparted via the impact of a flail or mace is transferred through the armour and into the target. Even if worn over padding such as a gambeson, maille provides only limited resistance against attacks from these weapons (Tihle 2017), and injuries can even be inflicted upon the head of a target through a substantial helmet (Mitchell 2002). These injuries can result in broken bones and/or cause significant internal bleeding depending upon the area of the body impacted, which would often result in prolonged, untreatable suffering with the medicine available in the middle ages (Mitchell 2002). The effectiveness of flails against

unarmoured (i.e., unhelmeted opponents) can be attested by remains found in the Holy Land, with impacts from spherical masses upon their skulls, sufficient to cause immediate death (Moreno 2015; Mitchell 2002). Due to the nature of the injuries they cause, flails can be utilised to stun or disable foes rather than outright kill them (Courville 1948), and this tactic was believed to be used by nomadic steppe tribes for the capture of prisoners (Kirpičnikov 1968; Osypenko 2020). The wounds from flails specifically are likely indistinguishable from those caused by similar blunt weapons such as maces (Geldof 2015).

The heads of most Eastern European flails tend to weigh between 2 and 300 g. With a combined handle, chain, and strap length of 70–80 cm (2–3 feet), a flail would strike, when swung at full force, with an equivalent impact of 6–14 kg. An 8 kg force is capable of breaking the strongest bones, though padded armour would provide some protection against this and increase the necessary impact required to disable the target. Against a more heavily armoured opponent, a strike to the head may serve to stun and thus temporarily disable them, even through a helmet (Kirpičnikov 1970).

Flails are highly effective weapons against unarmoured and maille-armoured targets (Lindsey 2005), but are markedly less useful against warriors equipped with plate armour, depending upon the design of the flail employed. As the "arms race" between armour penetration and protection advanced, so did the necessity to change the design of flails employed in warfare. As outlined above, maille, while being resistant to slashing attacks and to a degree thrusts, offers little in the way of protection against blunt force weapons such as maces and flails, due to its inherent flexibility. This weakness is amongst the reasons that maille eventually gave way to plate armour as metallurgy improved during the later 13th and 14th centuries (Carley 1962; Tihle 2017), also driven in large part by advances in missile weapon technology (Mäesalu 2004).

Compared to the majority of other basic weaponry, flails require a degree of specialism and some degree of skill in use (Hill 1998; Gallardo 2016), and would be rather situational as to their employment (Moreno 2015). Although the sceptics would cite the risk to one's own self and allies (Brooks 2019; Sturtevant 2016a, 2016b, 2017), the flail does possess a number of distinct advantages over other weaponry: to the unaware opponent, such a strange device would hold a potent psychological effect; the ability to entrance, hold off, or possibly even hit multiple opponents at the same time; bypass shields and blocks, and entrap opponents' weapons; and effectively negate the benefits of armour, all with minimal risk to the user if used properly (Moreno 2015). A lone flail would be of little benefit in an infantry battle line, with later medieval sources noting their effectiveness when used en masse, a tactic demonstrated to great effect by the Hussites (Grabarczyk 2000). One-handed flails were generally regarded as secondary weapons by infantry, used for exploitation once other blows had been struck with more conventional weapons such as spears and axes, or for close-in use by lighter-equipped troops such as horse archers (Kirpičnikov 1970; Sitdikov et al. 2015; Osypenko 2020).

Given the large variety of flail designs observed across Europe and the goals of this research, I shall not dwell too long on the construction of such weapons, but some overall conclusions can be drawn. The flexible portion (chain, rope, or thonging) of two-handed flails is often shorter than the haft, serving to increase the accuracy of blows and safety to the bearer at the expense of some damage potential (Waldman 2005). As with many other "blunt" weapons, flails tend to be somewhat more top-heavy than swords, though given the inherently chaotic nature of their motion, balance is secondary to the overall control and damage potential afforded to the user (Tihle 2017). Small flails were even made for children, to prepare them in the use of such weapons on the battlefield (Kirpičnikov 1970). The variation in the level of decoration on flail heads is significant (Kotowicz 2008); more basic (plain) flails are believed to have been the purview of the lower classes—more mass-produced weapons, while their more ornate counterparts were owned by the ruling classes (Kirpičnikov 1968). For example, a find from Poland depicts symbols commonly associated with Duke Oleg Swiatoslawicz (1055–1115) (Kotowicz 2008).

While in the modern era we may take a somewhat grim view of the nature of the medieval era, we must consider that, as we are, medieval people were products of their times. Despite their limited scientific knowledge, superstitions (Gordon 2019), and deeply held religious beliefs (Ames 2012), the vast majority of medieval people were practical and pragmatic, and would have likely recognised flails for what they were, highly effective—if somewhat chaotic and challenging to wield—weapons: a technology to be copied and adopted when encountered in other cultures as has occurred with many other concepts over the centuries. While I have not identified any texts to this effect, the common depictions of flails as weapons of demons, heretics, and pagans may reflect and parallel the similar held period beliefs around weapons such as crossbows—these were considered un-chivalric and thus banned by the Second Lateran Council (1139, Canon 29) (van der Veen 2012):

> "We prohibit under anathema that murderous art of *ballistariorum et sagittariorum*, which is hateful to God, to be employed against Christians and Catholics from now on"

In this context, the terms *ballistariorum* and *sagittariorum* were translated to mean "artillerymen or slingers" and "archers" respectively, though umbrella includes the crossbow as well.

## 10. Discussion, Conclusions, and Further Work

The breadth and depth of the sources outlined here present an interesting conundrum and, in some respects, raise more questions than they answer. I can unequivocally demonstrate that the peoples of Western, Central, and Southern early and high medieval Europe certainly had an awareness of flails as a general concept, at least as far back as 1100 and probably before this, based upon the many and varied references in the Matters of France and England, which draw upon earlier tales from the time of Charlemagne and before. The primary questions which arise from this are extent to which flails were used in battle, and how, when, and by whom the technology was introduced to the continent.

For the former of these, I would speculate that flail use outside of Eastern Europe was sporadic, infrequent, and situational, but not unknown, especially by the latter half of the 12th and into the 13th centuries, after which point artistic depictions are far more common and use more commonly cited in non-fictional texts. The combination of archaeological finds and other evidence places flails as extant technology within the period and region of interest, though I believe many of the various items previously identified as "weights" need to be reassessed in comparison with the Eastern European typologies of *kisten* heads. Threshing flails were definitely utilised as improvised weapons by the lower classes when called upon, even in spite of period laws such as the various "assizes of arms" and similar requiring the lower classes to possess actual weaponry and defensive tools such as spears and shields respectively (Hegg 2021; Hosler 2014). The evolution of threshing flails into the weaponised derivatives of the 14th century and onwards is well characterised and not in doubt.

For the latter question, there are a number of possibilities, but I would personally err towards an earlier introduction of flails with a low-level maintenance of the technology rather than a later adoption via contact with the Rus, through Byzantium or Poland, and so on in the 10th–11th centuries. Unless flails remained a constant technology from Roman times, the various invasions by the Huns (4th–6th centuries), Moors (8th–9th centuries), and Magyars (9th–10th centuries), to name a few, are the most likely sources. Given the extent of Frankish and Carolingian references to flails (from the Matter of France), the Huns and/or the Moors are the most likely sources from a French (and by extension English), Spanish, and Germanic perspective, while Byzantium and the Moorish invasion of Southern Italy (Sicily) likely provide the most appropriate explanation for Italian perspectives. Byzantium possibly retained flails as a technology from their Thessalian inception into the Roman era and onwards into the Middle Ages, or more likely rediscovered them in their many clashes with nomadic steppe peoples.

While I feel this work has presented a good basis for future study of flails in the early and high medieval period, I believe there is significantly more work which can be undertaken to both broaden and deepen our understanding. For example, there are few references to flails being employed in the Nordic lands outside of Finland, and I would welcome any further knowledge which would expand the library of sources for that area of the world, alongside more Germanic, Iberian, and Italian depictions and descriptions. Although several comprehensive texts discussing the evolution of weaponry in some of these areas of interest are available, they do not appreciably discuss flails, if at all, and miss several of the well-known iconographical sources which depict the weapons of interest to this work (Riquer 1968; Soler del Campo 1985). I would especially welcome the highlighting of archaeological finds from the area of interest which resemble flails heads, and any further artistic or literary depictions and descriptions beyond those presented here, given the challenges incurred with language barriers, travel limitations, and so on.

**Funding:** This research received no external funding. The APC was waived for this work.

**Institutional Review Board Statement:** Not applicable.

**Informed Consent Statement:** Not applicable.

**Data Availability Statement:** Not applicable.

**Acknowledgments:** I wish to thank N. Dos Reis (University of Poitiers, CESCM) for his valuable discussions and help with the French texts and translations, S. Gordon (University of Cardiff) and S. Bennett, for their valuable discussions and proofing, B. Peart for his kind proofing, G. Caulfield (UCL) for her valuable discussions and the suggestion of the title, P. Alekseychik (Medieval Advisor) for originally inspiring this work and his tireless presentation of eclectic and poorly known medieval sources, M. H. López for her generous assistance with the Catalan and Spanish translations, T. Vlasatý (Project Forlǫg) for providing a few key references from the Central and Eastern European side of things and some helpful discussions, the various members of the Historia Normannis, Sericum et Gladium, and Chanz des Reis historical re-enactment societies for their valuable discussions in the preparation of this work, and I am especially indebted to the staff at various national and academic libraries for their assistance in sourcing images, manuscript information, and associated permissions. All images are © the original owners and are available online publicly, with links provided where possible and copyright permissions noted in the captions of each image. Although this work is published under the CC-BY 4.0 license, some images are reproduced under other licenses as noted in their captions where applicable.

**Conflicts of Interest:** The author declares no conflict of interest.

## Appendix A

Here I present a selection of miscellaneous supporting images including some late-medieval folios from illuminated manuscripts (Figure A1) and other artistic depictions (woodcuts) (Figure A2), several archaeological finds which may be flail heads that have been misidentified (Figures A3–A5), and links to flails and their components which have been sold at auction in recent years.

*Appendix A.1. Manuscript Images*

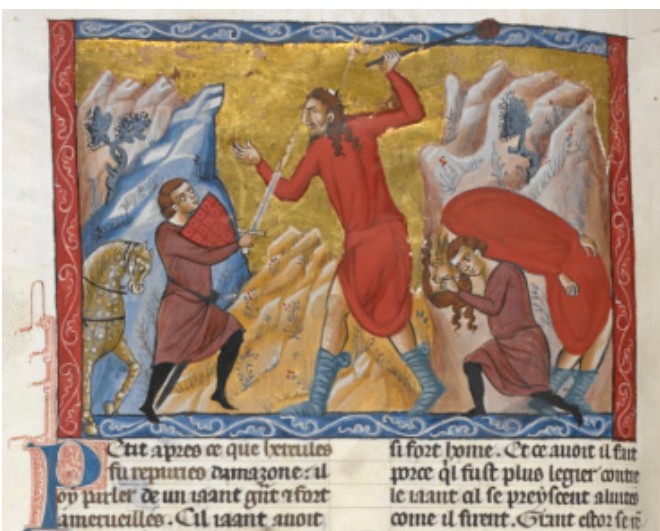

**Figure A1.** Fol. 104v (battle between Hercules and Antaeus) from *Historia Ancienne Jusqu'a César*, Acre, 1275–1291. (© British Library Board, BL MS 15268, permission granted for non-commercial reproduction, Image from http://www.bl.uk/manuscripts/FullDisplay.aspx?ref=Add_MS_15268&index=0, accessed on 25 October 2023).

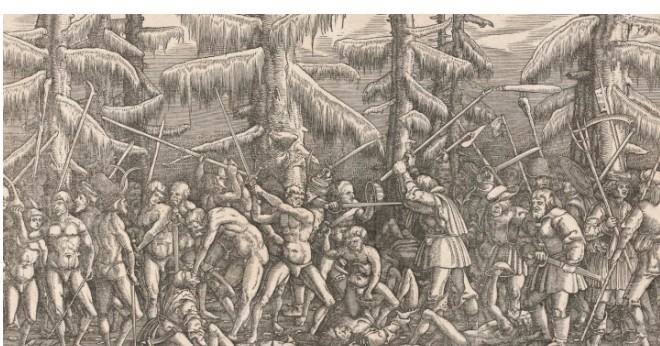

**Figure A2.** "The Battle of Naked Men and Peasants", Hans Lützelburger (dated 1522). (Image in the public domain, provided courtesy of US National Gallery of Art, Washington, USA, from https://www.nga.gov/collection/art-object-page.208472.html, accessed on 25 October 2023).

*Appendix A.2. Possibly Misidentified Archaeological Finds from the UK*

The UK Potable Antiques Scheme has a large database of finds. Searching using the term "steelyard weight" reveals a great many artefacts recovered which match the pattern of flail heads from Eastern Europe. Given the weights of these do not often match up with the ounces used at the time, they may well be misidentified flail heads. I am unaware of any similar artefact databases from other countries in the region of interest and so would welcome direction to or access to these in order to aid this research. A small selection from the UK database within the likely mass range for flail heads (1–200 g) is presented below for context.

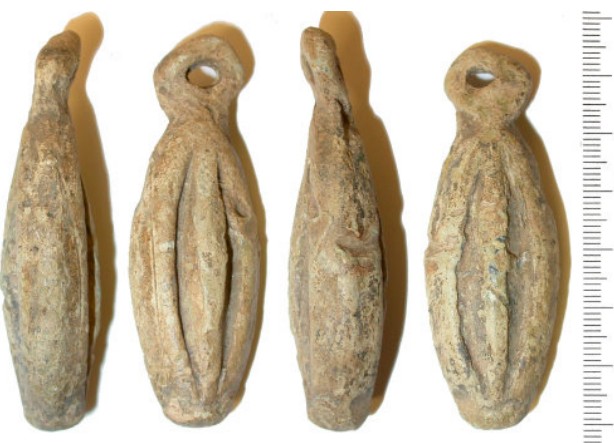

**Figure A3.** Lead alloy weight (Item DENO-0A3621) from W. Midlands, possibly a flail head, matching E. European patterns, dated 1066–1800. Weighs 90 g. (Image Courtesy of the Portable Antiques Scheme, shared under CC BY 3.0 DEED license, available with more information at https://finds.org.uk/database/artefacts/record/id/283924, accessed on 25 October 2023).

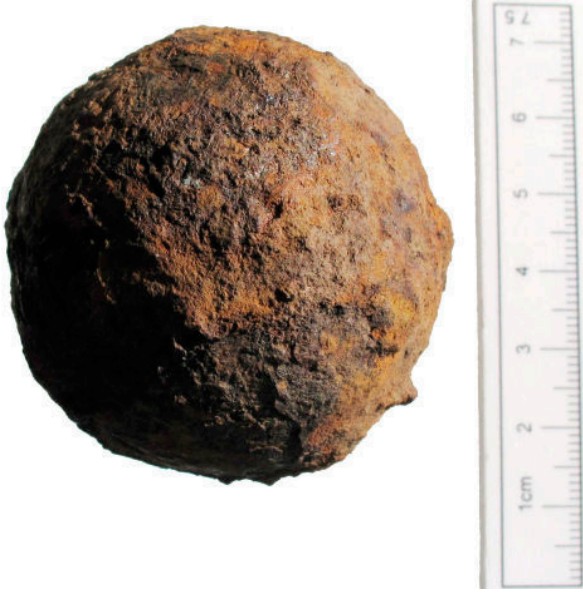

**Figure A4.** Possibly morning star (or flail) head (Item NCL-A14BD8) from NE England. Spikes have corroded over time. Dated 1300–1600. Weighs 51 g. (Image Courtesy of the Portable Antiques Scheme, shared under CC BY 3.0 DEED license, available with more information at https://finds.org.uk/database/artefacts/record/id/516336, accessed on 25 October 2023).

*Appendix A.3. Finds Sold at Auction*

A selection of flail heads, sometimes with attached chains, appear at antique auctions from time to time. These are often dated within a wide window of provenance, and as such do not represent the most reliable source of information, but are in some cases representative of the range of flail construction observed across the Middle Ages, though some items may be reproductions and thus not be genuine. A few examples are presented in https://www.thelanesarmoury.co.uk/shop.php?d=1&q=flail, https://www.npcollectables.org.uk/a1-gallery3.asp?roomID=1027&pictureID=97382, https://www.liveauctioneers.com/en-gb/price-result/medieval-iron-flail-with-chain/, and https://www.worthpoint.com/worthopedia/ancient-viking-german-kisten-ball-1917686028 for reader interest, all of these were accessed on 25 October 2023.

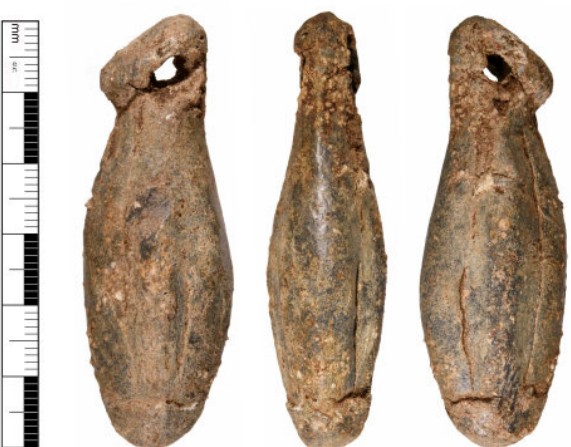

**Figure A5.** Lead alloy weight (Item PUBLIC-AF81CE) found in Norfolk, dated 1300–1700. Matches several Eastern European flail patterns. Weights 85.48 g. (Image Courtesy of the Portable Antiques Scheme, shared under CC BY 3.0 DEED license, available with more information at https://finds.org.uk/database/artefacts/record/id/1102234, accessed on 25 October 2023).

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
