# Peer review of "Fantastic Flails and Where to Find Them: The Body of Evidence for the Existence of Flails in the Early and High Medieval Eras in Western, Central, and Southern Europe"

_2409-9252, doi:10.3390/histories4010009_

Round 1
Reviewer 1 Report
Comments and Suggestions for Authors
The author analyzes the type of weapon called the flail, which has received scant attention in the medieval literature. The author collects a large number of literary, archaeological, and artistic sources to confirm that the flail was a type of weapon that was not only frequently used in Eastern Europe and Byzantium, but also in Western Europe. In this way, he concludes that this type of weapon was part of the medieval war culture of the entire continent and that its use was much more widespread throughout Europe than previously thought. This article is an excellent summary about the use of the flail in Medieval Europe.
Vevertherless, as the author recognizes, the study suffers from some gaps in the bibliographic references, due to the diversity of languages in which studies on medieval weaponry have been published. However, some important studies missing could be included, such as:
Riquer, M. de:, L'arnès del cavaller: armes i armadures catalanes medievals. Ariel, 1968
Soler del Campo, A., El armamento medieval en la Península Ibérica: siglos X y XI. Madrid, 1985.
Soler del Campo, A., La evolución del armamento medieval en el reino castellano-leonés y al-Andalus (Siglos XII-XIV). Madrid, 1993;
Kuleshov Yu.A. "Combat Flails in the Armament of the Golden Horde". Zolotoordynskoe obozrenie=Golden Horde Review. 2019, vol. 7, no. 1, pp. 37–54. / Kuleshov, Yuriĭ Alekseevich. “Боевые цепы в комплексе вооружения Золотой Орды.” Zolotoordȳnskoe obozrenie / Golden Horde Review, vol. 7, no. 1, 2019, pp. 37-54.
There are some minor errors in the titles of the references, such as the first one, whose correct title is: Atlas ilustrado de la Guerra en el Edad media en España.
Author Response
I thank the reviewer for taking the time to look over my sizeable manuscript and their compliments and comments on the work.
I have reviewed the references you suggested that were available and accessible (Riquer (physical copy in UoM library), Soler del Campo #2 (interlibrary-loan from Cambridge), and Kuleshov (available online); the first Soler del Campo text suggested was unavailable).
With the translation help of several Catalan-fluent colleagues, I could not find any reference to flails in the 1968 work of Riquer - although the word “maça” is present in the index, this is not directly mentioned in the pages stated therein, only “porra”, perhaps in reference to a club, or similar bludgeon? There are similarly no references in this text to the terms “maça amb cadenes", “batolla” or the more common Spanish term “mangual” used to describe these kinds of weapons.
Similarly, while the comprehensive 1985 work of Soler de Campo has a great many excellent descriptions and depictions of swords, lances, helmets, maille/plate armours, and shields with associated typologies, and a brief investigation of maces (as mazas) I can, with the assistance of Spanish-fluent colleagues, find no reference to flails under the terms of “maza/mazo con/de cadena”, “mangual”, “rompecabezas”, “látigo de armas”, “látigo de guerra", or “mayal de armas” in this text. This is not aided by the absence of an index within the text, though at least the common weapons (sword, lance, and mace) are grouped by chapter. The famous depiction of a flail from the arch of the church of San Miguel de Uncastillo is likewise absent from this text, suggesting to me that flails were not a focus for the author. The introduction and conclusions of this work nonetheless provided a useful insight into the context of historiological weapons research which has been highlighted and this text cited as a result.
The Kuleshov text provided useful context regarding flail use by the Mongols and has been added to the text and cited accordingly.
I have corrected the reference title as suggested and checked the remainder for errors.
Reviewer 2 Report
Comments and Suggestions for Authors
Only a few reservations can be made regarding this very valuable study:
1. Before publishing the text, the author should try to read the texts of written sources that he could not use in the original. There are several such cases (e.g. "Servaes Legende" of Heinrich von Veldecke or "Perceval" of Chretien de Troyes). Detailed analysis must be based on the original source text.
2. Minor typographical errors in the records of Slavic (especially Polish) names and surnames have been found: Niewinski (verses 218, 2003) - should be: Niewiński; Gorski, Gorsky (v. 447, 451, 453, 462-3, 1841) - should be: Górski; Wilczynska (v. 447, 451, 453, 462, 464, 1841) - should be: Wilczyńska; Radoslaw (v. 1888) - should be: Radosław; Skowronski (Pawel) (v. 572, 627, 1928) - should be: Skowroński (Paweł); Lawrynowicz (v. 446, 1935) - should be: Ławrynowicz; Tyniev, A. (v. 2042) - schould be: Tyniec, Anna.
Author Response
I thank the reviewer for taking the time to look over my sizeable manuscript and their compliments and comments on the work. In response to the suggestions made:
1) At your suggestion I have revisited the written sources cited, including those highlighted, including a deeper search for copies of the original texts or more preferably, modern English, French, or German translations. The text has been clarified in several places to address the nature of the translations from which my information was sourced, with several newly-identified complete texts added to the citation list. I have further clarified the quotes taken with line or verse numbers where these are available. Some additional discussion has been added where these can be drawn from original texts.
2) I have corrected the Slavic names as suggested.